# Antifungal Properties of Bio-AgNPs against *D. pinodes* and *F. avenaceum* Infection of Pea (*Pisum sativum* L.) Seedlings

**DOI:** 10.3390/ijms25084525

**Published:** 2024-04-20

**Authors:** Karolina Stałanowska, Joanna Szablińska-Piernik, Agnieszka Pszczółkowska, Viorica Railean, Miłosz Wasicki, Paweł Pomastowski, Lesław Bernard Lahuta, Adam Okorski

**Affiliations:** 1Department of Plant Physiology, Genetics and Biotechnology, University of Warmia and Mazury in Olsztyn, Oczapowskiego 1A, 10-719 Olsztyn, Poland; karolina.stalanowska@uwm.edu.pl (K.S.); lahuta@uwm.edu.pl (L.B.L.); 2Department of Botany and Evolutionary Ecology, University of Warmia and Mazury in Olsztyn, Pl. Łódzki 1, 10-719 Olsztyn, Poland; joanna.szablinska@uwm.edu.pl; 3Department of Entomology, Phytopathology and Molecular Diagnostics, University of Warmia and Mazury in Olsztyn, Pl. Łódzki 5, 10-727 Olsztyn, Poland; agnieszka.pszczolkowska@uwm.edu.pl; 4Department of Infectious, Invasive Diseases and Veterinary Administration, Institute of Veterinary Medicine, Nicolaus Copernicus University in Toruń, Gagarina 7, 87-100 Toruń, Poland; viorica.railean@umk.pl; 5Centre for Modern Interdisciplinary Technologies, Nicolaus Copernicus University in Toruń, Wileńska 4, 87-100 Toruń, Poland; milos13w@outlook.com (M.W.); p.pomastowski@umk.pl (P.P.); 6Department of Inorganic and Coordination Chemistry, Nicolaus Copernicus University in Toruń, Gagarina 7, 87-100 Toruń, Poland

**Keywords:** AgNPs, nanoparticles, *Didymella pinodes*, *Fusarium avenaceum*, *Pisum sativum*, fungal diseases, plant protection, pea seedlings, metabolic profiles

## Abstract

Ascochyta blight and Fusarium root rot are the most serious fungal diseases of pea, caused by *D. pinodes* and *F. avenaceum*, respectively. Due to the lack of fully resistant cultivars, we proposed the use of biologically synthesized silver nanoparticles (bio-AgNPs) as a novel protecting agent. In this study, we evaluated the antifungal properties and effectiveness of bio-AgNPs, in in vitro (poisoned food technique; resazurin assay) and in vivo (seedlings infection) experiments, against *D. pinodes* and *F. avenaceum*. Moreover, the effects of diseases on changes in the seedlings’ metabolic profiles were analyzed. The MIC for spores of both fungi was 125 mg/L, and bio-AgNPs at 200 mg/L most effectively inhibited the mycelium growth of *D. pinodes* and *F. avenaceum* (by 45 and 26%, respectively, measured on the 14th day of incubation). The treatment of seedlings with bio-AgNPs or fungicides before inoculation prevented the development of infection. Bio-AgNPs at concentrations of 200 mg/L for *D. pinodes* and 100 mg/L for *F. avenaceum* effectively inhibited infections’ spread. The comparison of changes in polar metabolites’ profiles revealed disturbances in carbon and nitrogen metabolism in pea seedlings by both pathogenic fungi. The involvement of bio-AgNPs in the mobilization of plant metabolism in response to fungal infection is also discussed.

## 1. Introduction

Field pea (*Pisum sativum* L.) is an important pulse because of its widespread use in human food and animal feed production, but it is also crucial for agriculture. Like other legumes, it can fix atmospheric nitrogen and make it more available to other plants, which reduces the global use of synthetic nitrogen fertilizers. Pea is also cultivated as a rotation and disease-breaking crop. These special attributes make pea important for both natural and agricultural ecosystems [1,2]. Pea seeds have high nutritional value owing to their high-quality protein content (15–30%), high amino acid availability (such as arginine, phenylalanine, leucine, isoleucine, tryptophan, lysine, and threonine), and energy (carbohydrate content 24–49%) and mineral element content (such as potassium, phosphorous, magnesium, and calcium) [3,4]. The main global producers of dry pea in 2022 were the Russian Federation (3.6 Mt), Canada (3.4 Mt), China (1.5 Mt), India (1 Mt), the USA (684 kt), Ethiopia (400 kt), France (400 kt), Germany (323 kt), Argentina (297 kt), and Australia (261 kt) [5]. The main limitation in the cultivation of peas is the lack of yield stability caused by abiotic and biotic stresses [6]. The most devastating fungal diseases of pea are Ascochyta blight and root rot [2,7]. 

Ascochyta blight, also known as “black spot” disease, is a major foliar pea disease that occurs in Europe and other countries, such as Canada, Australia, the USA, and China, and causes a loss that can reach 75% [2,8,9]. The disease causes dark-brownish necrotic lesions on the aboveground parts of the plant, including pods and seeds, which affect their quantity and quality [8,10]. Ascochyta blight of pea is caused by a fungal complex that includes *Didymella pinodes* (formerly known as *Mycosphaerella pinodes*), *Didymella pisi*, and *Didymella pinodella* (also known as *Ascochyta pinodella* and *Phoma medicaginis* var. *pinodella*) [8,10,11]. In Australia, the disease is also associated with *Ascochyta koolunga* (also known as *Phoma koolunga*) [12,13]. However, the occurrence of this pathogen has not been reported elsewhere in the world [14]. Ascochyta blight pathogens can coexist as a fungal complex or infect plants individually, and their prevalence varies worldwide. However, *D. pinodes* is considered to be the most pathogenic and damaging pathogen associated with the disease, regardless of the region-dominant species [2,8]. Moreover, *D. pinodes* is also associated with pea root rot [15]. Pea root rot is a complex disease caused by fungal and oomycete pathogens including *Fusarium* spp., *Pythium* spp., *Phytophthora* spp., *Rhizoctonia* spp., *Didymella* spp., *Aphanomyces euteiches*, and *Thielaviopsis basicola* [16]. The disease can reduce the pea crop yield by 30% to 57% [17]. Pea root rot is sometimes referred to as Fusarium root rot as *Fusarium* species are the predominant pathogens [16,18]). Typical symptoms of root rot are dark-brown lesions on the roots (especially at the base of the roots, close to the soil level); dark spots on the cotyledons, epicotyl, or base of the shoot; and leaf yellowing [19]. The primary causal agents are *Fusarium avenaceum*, *F. solani*, and *F. oxysporum*. The predominance of each species varies by year and climate conditions [7,20,21]. In a temperate climate, the most prevalent species is *F. aveanceum*, which has been reported in Europe [22,23], Canada [20,24] and the USA [21]. 

*Didymella pinodes* and *Fusarium avenaceum* are becoming increasingly problematic as pea pathogens. Both are seed-borne pathogens (*F. avenaceum* is also a soil-borne pathogen) and can overwinter in the soil or on plant debris [8,25]. Another challenge is the wide host ranges of *D. pinodes* (mainly legumes) [9,26] and *F. avenaceum* (legumes, cereals, and other perennial plant species) [25,27], which can serve as vectors for the spread of pathogens. To control and reduce the severity of pea diseases, many agronomic and physiological practices have been implemented, including crop rotation, intercropping, the burial of infected debris, and fungicide use [7,9]. However, fungicide application shows only a moderate effect on root rot [27]. Breeding a resistant pea cultivar is considered to be the most promising approach for reducing susceptibility to diseases [6]. However, pea cultivars fully resistant to Ascochyta blight or root rot have not yet been bred [9,16,27,28,29]. Therefore, temporary solutions are required until new resistant cultivars are established. In pea protection against Ascochyta blight and Pea root rot, various fungicides are used, with active substances such as strobilurins (azoxystrobin, pyraclostrobin); carboxamides (boscalid, bixafen), which affect respiration as quinone outside inhibitors (QoI) and succinate-dehydrogenase inhibitors (SDHI), respectively; triazoles (tebuconazole, prothioconazole, difenoconazole), which inhibit sterol biosynthesis in membranes as demethylation inhibitors (DMI); or phenylpyrroles (fludioxonil), which affect signal transduction, used most frequently [30,31,32,33,34]. The use of fungicides is gradually becoming limited because of the fungicide resistance of pathogens and their hazardous effects on the environment and human health [35,36]. Therefore, alternative solutions are needed. 

Metal nanoparticles are becoming more popular and are widely considered to be plant protection agents [36,37] or additives to the currently used fungicides to improve their effectiveness [36]. The antimicrobial properties of metal nanoparticles are widely known, and their applications in plant disease management have been intensively investigated [37,38]. The size distribution, shape, composition, crystallinity, agglomeration, coating surface, and charge of the nanoparticles underlie their antifungal activity. Nanoparticles can affect fungi at various levels. They can interact with membrane proteins, alter membrane potential and electron transport, and induce the generation of reactive oxygen species (ROS) and protein oxidation, which leads to cell membrane permeability and integrity disruption, DNA damage, changes in gene expression, the suppression of spores’ germination and development, and, finally, cells death and mycelial destruction [37,39]. The complex mechanisms of action of nanoparticles suggest that there is a small risk of fungi developing resistance to them, though certainly not to the same extent as in the case of currently used fungicides [39].

Silver nanoparticles (AgNPs) are one of the most frequently and extensively investigated nanoparticles. The fungistatic properties of AgNPs have been evaluated in vitro against some *Fusarium* species, such as *F. graminearum* [40,41,42,43], *F. culmorum* [40,41,43], *F. oxysporum* f. sp. *radicis-lycopersici* [44,45], *F. oxysporum* [40,41], *F. equiseti* [41,46], *F. langsethiae* [41], *F. poae*, *F. proliferatum*, *F. sporotrichioides*, *F. verticillioides*, [40,41], *F. solani* [44], and *F. avenaceum* [40,43,46]. To date, there have been no investigations examining the applications of AgNPs either in vitro, in vivo, or in greenhouse or field experiments; against *D. pinodes* and for *F. avenaceum*, there is only limited information available from in vitro experiments [40,43,46]. 

The method of silver nanoparticle synthesis affects their physicochemical properties and antimicrobial activity. Physical and chemical methods of AgNP synthesis are expensive, and hazardous chemicals are used. Biological synthesis, also called “green synthesis”, is more environmentally friendly, cost-effective, and sustainable. This approach is based on the use of bacteria, fungi (including yeast), plants and their extracts, enzymes, and biomolecules. Bacteria are widely used in NP production because of their high yield, low growth condition requirements, and ease of purification. Moreover, they can synthesize nanoparticles extracellularly or intracellularly [47,48]. Many studies have confirmed that biologically synthesized metal nanoparticles are less toxic to plants and animals than chemically synthesized ones [49,50,51,52], and at the same time, they exhibit stronger antimicrobial properties [53,54]. Up to now, the antimicrobial properties of bio-AgNPs, produced using *Lactobacillus paracasei* isolated from whey, against various dairy products’ foodborne pathogens and both drug-resistant Gram-negative (*Proteus mirabilis*, *Acinetobacter baumannii*, *Pseudomonas aeruginosa*) and Gram-positive bacteria (*Staphylococcus epidermidis* and *Micrococcus luteus*) have been documented [55]. However, their toxicity against phytopathogenic fungi remains unknown. Therefore, this study aimed to evaluate the antifungal/fungistatic properties and effectiveness of bio-AgNPs, in in vitro (poisoned food technique; resazurin assay) and in vivo (seedlings infection) experiments, against important fungal pathogens of pea (*Pisum sativum* L.)—*Didymella pinodes* and *Fusarium avenaceum*. Moreover, the effect of seedling infection on the further metabolic profiles of the roots, shoots, and cotyledons was analyzed. The comparison of changes in the concentrations of polar metabolites revealed a strong disruption of carbon and nitrogen metabolism by both pathogenic fungi. 

## 2. Results

### 2.1. Antifungal Properties of Bio-AgNPs In Vitro

The antifungal properties of bio-AgNPs against *D. pinodes* and *F. avenaceum* were evaluated using a resazurin assay to determine the minimum inhibitory concentration (MIC) of bio-AgNPs against fungal spores and using the poisoned media technique to investigate mycelial growth inhibition. After the serial dilution of bio-AgNPs, the MIC for both *D. pinodes* and *F. avenaceum* spores was 125 mg/L (the lowest concentration without resazurin discoloration compared with the control without spores; Figure 1A,B). 

The inhibitory effects of bio-AgNPs on the linear mycelial growth of both fungi were also observed (Figure 2). They increased with increasing concentrations of bio-AgNPs (Figure 2, Table 1). The highest inhibition of *D. pinodes* was observed in bio-AgNPs at a concentration of 200 mg/L—mycelial growth inhibition (MGI) was 49% and 45% after 7 and 14 days of incubation, respectively (Figure 2A,B). The lowest tested concentration of nanoparticles slightly stimulated the mycelial growth of *F. avenaceum* (Figure 2C,D, Table 1). Bio-AgNPs at concentrations of 100–200 mg/L inhibited the growth of *F. avenaceum* at similar levels, and the MGI was approximately 55% and 25% after 7 and 14 days of incubation, respectively, at the highest tested concentration (Table 1). However, bio-AgNPs, even at the highest tested concentration, were significantly less effective than the fungicides. Based on these results, we decided to test the in vivo fungistatic activities of bio-AgNPs at concentrations of 100 and 200 mg/L. 

### 2.2. Pea Seedlings Infection and Antifungal Properties of Bio-AgNPs In Vivo

Symptoms of *D. pinodes* infection (dark spots) were found on the cotyledons at the base of the shoots and roots. Moreover, tendril and stipule wilting were also noted (Figure 3B). 

The infection inhibited the elongation of roots and the FWs of roots and shoots. The short-term immersion of seedlings in fungicide and bio-AgNPs reduced this negative effect—seedlings’ FWs and root lengths were as high as in control seedlings (Table 2). 

The symptoms of *F. avenaceum*, such as dark-brownish spots or necrotic changes (dependent on the level of infection), were observed on the roots and cotyledons (Figure 4(B1,B2)).

Deleterious effects were also found on shoots—the shortening of the stem and underdevelopment of stipules and tendrils (Table 2, Figure 4B). In effect, the FWs and lengths of shoots considerably decreased. However, those negative effects were alleviated by seedlings’ pre-treatment with a fungicide or bio-AgNPs (Figure 4C–E, Table 2). No significant changes were observed in the roots’ FWs and lengths (Table 2) and the DWs of roots and cotyledons (Appendix A).

The disease index (DI) of the water-pre-treated seedlings infected both with *D. pinodes* and *F. avenaceum* was approximately 30% (Figure 5A,B). In both cases, the seedlings’ pre-treatment with a fungicide and bio-AgNPs significantly reduced the DI to a similar extent—to less than 5% and 7% in the cases of *D. pinodes* and *F. avenaceum* infection, respectively. The level of actual infection, considered to be the level of fungal gDNA in seedlings’ tissues (analyzed via qPCR), differed between those two pathogens. However, in both cases, bio-AgNPs and the fungicide significantly reduced the level of infection, as well as the DI. *D. pinodes*’ gDNA quantity was almost 30 pg in infected seedlings after short-term immersion in water, and fungicide and bio-AgNP pre-treatments decreased its level to 0.28–0.58 pg (Figure 5C).

The level of *F. avenaceum* gDNA was similar in infected seedlings pre-treated with fungicide and 100 mg/L of bio-AgNPs (46 and 50 pg, respectively), whereas in seedlings treated with 200 mg/L of bio-AgNPs, it was twice as high (104 pg) but still significantly lower than that in infected-water-pre-treated seedlings (507 pg) (Figure 5D). 

A strong positive correlation between the quantity of fungal gDNA in seedlings tissues and their disease index was found—R = 0.94 and R = 0.98 (*p* < 0.001) for *D. pinodes* (Figure 6A) and *F. avenaceum* (Figure 6B), respectively.

### 2.3. Polar Metabolic Profiles 

#### 2.3.1. Metabolic Profiles of Control Seedlings

In the tissues of 22-day-old pea seedlings, 31 metabolites were identified and classified into soluble carbohydrates (fructose, galactose, glucose, *myo*-inositol, sucrose, and gluconic acid), amino acids (alanine, asparagine, aspartic acid, β-alanine, γ-aminobutyric acid (GABA), glutamic acid, homoserine, hydroxyproline, isoleucine, lysine, phenylalanine, proline, serine, threonine, tyrosine, and valine), organic acids (butyric acid, citric acid, lactic acid, malic acid, malonic acid, oxalic acid, and succinic acid), and the remaining compounds (phosphoric acid and urea; Appendix A). Although the concentration of total identified polar metabolites (TIPMs), including sub-fractions, in the control seedlings in the experiment with *D. pinodes* (Appendix A) was higher than that in the experiment with *F. avenaceum* (Appendix A), the overall metabolic profiles were alike. In both experiments, the concentrations of TIPMs in control seedlings were much higher in cotyledons (189.84 and 157.92 mg/g DW) than in shoots (121.26 and 117.80 mg/g DW) and roots (64.90 and 59.52 mg/g DW, respectively). Moreover, the major fraction of polar metabolites was soluble carbohydrates, sharing up to 74–76% of TIPMs in cotyledons, whereas amino acids dominated in roots and shoots (approximately 60 and 50% of TIPMs, respectively). The major amino acids, regardless of the tissues analyzed, were homoserine and asparagine, whereas sucrose was the major carbohydrate (in all tissues) and, among organic acids, dominated citrate (in cotyledons and shoots) or malate (in roots; Appendix A). 

#### 2.3.2. Metabolic Profiles of Infected Seedlings

##### PCA

The principal component analyses (PCA) of the polar metabolites of the roots and shoots of both control and infected seedlings revealed different data distributions (Figure 7).

The roots’ samples of control seedlings and those infected with *D. pinodes* after pre-treatment with fungicide were grouped at the right site (bottom and top, respectively), according to PC1, sharing 75.16% of the variability (Figure 7A). Control samples of shoots were also clearly separated from the samples of *D. pinodes*-infected seedlings (which were placed on the left side of the plot), according to PC1, sharing 94.86% of the variability (Figure 7B). Moreover, shoots’ samples of seedlings pre-treated with water were grouped in the left top corner. The samples of roots and shoots of *F. avenaceum*-infected seedlings (Figure 7C and Figure 7D, respectively) were separated from the samples of control seedlings and those pre-treated with water, according to PC1 (for roots) or PC2 (for shoots), sharing 84.28 and 27.98% of the variability, respectively. 

According to the PCA loading plots, the distribution of the roots’ samples was mainly related (in both experiments) to differences in the concentrations of homoserine, asparagine, sucrose, glucose, phosphoric acids, and, additionally, GABA (in *D. pinodes*-infected seedlings, Figure 7E) or malate (in *F. avenaceum*-infected seedlings, Figure 7G). The distribution of shoots’ samples was mainly related to sucrose, homoserine, asparagine, and, additionally, phosphoric acid (samples of *D. pinodes*-infected seedlings, Figure 7F) or proline (samples of *F. avenaceum* infected seedlings, Figure 7H). 

The cotyledons samples’ distribution according to the PCA varied between infections with different pathogens. (Appendix A). The samples of cotyledons of control and *D. pinodes*-infected seedlings were clustered in the upper right corner of the PCA plot (according to PC1 and PC2, sharing 88.9 and 5.99% of variance, respectively). This distribution was determined by differences in the contents of sucrose, homoserine, and asparagine. In turn, samples of cotyledons from seedlings infected with *F. avenaceum*, pre-treated with bio-AgNPs before infection, were separated from those treated with a fungicide and water (according to PC1, accounting for as much as 97.93% of the variance). This sample distribution was influenced by the same set of metabolites plus glucose and fructose. 

##### Changes in the Concentrations of Metabolites

The inoculation of pea seedlings with *D. pinodes* or *F. avenaceum* spores, without seedlings’ pre-treatment with a fungicide or AgNPs, caused not only the development of diseases but also significant changes in the composition and concentration of polar metabolites. Compared to the control seedlings, the roots of infected seedlings contained less soluble carbohydrates and more amino acids and organic acids, while in the shoots, the concentrations of carbohydrates and amino acids dramatically decreased (Table 3).

The short-term immersion of seedlings in a fungicide or bio-AgNPs, causing seedlings’ protection against the development of both pathogens (Table 2, Figure 3, Figure 4, Figure 5 and Figure 6), also affected the concentration of polar metabolites. In seedlings pre-treated with fungicide and further infected with *D. pinodes*, TIPMs were significantly (*p* < 0.05) higher in roots, while lower in shoots, compared with control seedlings and those pre-treated with water. Moreover, seedling pre-treatment with bio-AgNPs resulted in a reduction in TIPMs in both roots and shoots (Table 3).

In the roots of seedlings pre-treated with a fungicide and bio-AgNPs and infected with *F. avenaceum*, TIPMs were significantly elevated, compared with the control and those pre-treated with water, due to elevated levels of all sub-fractions (Table 3). In shoots, however, TIPMs were lower (in fungicide-pre-treated samples), elevated (in bio-AgNPs at 100 mg/L), or remained unchanged (in bio-AgNPs at 200 mg/L). The above differences in the TIPMs (and sub-fractions) resulted from changes in most metabolites (Appendix A), i.e., these dominant compounds: sucrose (among sugars) and homoserine, asparagine, and GABA (among amino acids; Figure 8).

The changes in polar metabolites in the cotyledons of seedlings infected with *D. pinodes* differed from those in the cotyledons of seedlings infected with *F. avenaceum* (Appendix A, respectively). The concentration of TIPMs in the cotyledons of seedlings infected with *D. pinodes* was significantly higher than those in the other seedlings (except for those previously treated with the fungicide; Appendix A). In these cotyledons, the concentrations of sucrose, succinic acid, and several amino acids (asparagine, isoleucine, phenylalanine, GABA) were significantly elevated. In the cotyledons of *F. avenaceum*-infected seedlings, significantly increased TIPMs were detected in bio-AgNP-pre-treated seedlings, which resulted from the accumulation of sucrose, *myo*-inositol, most amino acids, and phosphoric acid (Appendix A).

## 3. Discussion

### 3.1. Antifungal Activity of Bio-AgNPs against D. pinodes and F. avenaceum

In the present study, we investigated the antifungal properties of bio-AgNPs against selected phytopathogenic fungi, both in in vitro and in vivo in pea seedlings. To the best of the authors’ knowledge, this is the first report of the application of metal nanoparticles to control pea infection caused by *D. pinodes* and *F. avenaceum.* Quantitative analysis revealed that the elemental silver content in the bio-synthesized silver nanoparticles (bio-AgNPs) was found to be 9.03 ± 0.02 mg/mL. X-ray Photoelectron Spectroscopy (XPS) was employed to examine the surface characteristics of the nanoparticles, confirming the presence of metallic silver. The observed doublet for Ag(3d 5/2) and Ag(3d 3/2) at binding energies of 368.1 eV and 374.1 eV, respectively, as shown in Appendix A, and the energy splitting of 6 eV between these peaks are characteristic of metallic silver, thereby corroborating the formation of a metallic state in the nanoparticles. Further analysis using XPS revealed that the total silver content in the nanoparticles was 11.25 ± 0.02% by atomic percentage. Notably, the majority of the silver was in the metallic state (Ag(0)), accounting for 98.9 ± 0.1% of the total silver, while the ionic form (Ag(+)) constituted only 1.1 ± 0.1%. This predominance of metallic silver suggests a high reduction efficiency during the synthesis process, which is critical for applications where the antimicrobial properties of metallic silver are desired. These findings align with previous studies, such as the research conducted by Kim et al. [56], which emphasized the role of metallic silver grains in biological processes. The investigation into the elemental composition of bio-AgNPs, as shown in Appendix A, confirmed the presence of silver within the range of 2.7–3.2 keV, which aligns with the expected binding energies for metallic silver. The signals corresponding to carbon, oxygen, phosphorus, and sulfur indicate that these elements comprise the organic surface coating of the nanoparticles. This coating likely plays a crucial role in the stability and bioactivity of the nanoparticles, as suggested in the study by Railean et al. [57], which proposed that organic constituents in bio-AgNPs can significantly influence their formation and functional properties. In Appendix A, the predominant population of bio-AgNPs is shown to be spherical and homogeneous, with an average size of approximately 18.25 ± 0.58 nm, closely centering around 20 nm. The physicochemical properties of bio-AgNPs, characterized through Dynamic Light Scattering (DLS) (scattering angle at 173°) [55,57] and Transmission Electron Microscopy (TEM), indicate a stable colloidal system with a hydrodynamic diameter of 100 to 150 nm and a polydispersity index (PDI) below 0.7, suggesting minimal aggregation and enhanced stability. The zeta potential values, ranging from −23 to −41 mV, further confirm the stability, which is critical for biological applications where nanoparticle aggregation could impede their functionality [58]. In agricultural contexts, such as combating *D. pinodes* and *F. avenaceum* infections in pea seedlings, the stability and uniformity of bio-AgNPs contribute to their efficacy. The physicochemical properties of nanoparticles, such as size distribution, shape, composition, crystallinity, agglomeration, coating surface, and charge, affect their antifungal properties [59]. 

The fungistatic effect of nanoparticles might be difficult to compare between different studies because of the nanoparticles’ properties, the fungal species or strains used, and the different methodologies used for evaluating fungicidal/fungistatic properties. Fungal spores are usually more sensitive to nanoparticles than mycelium hyphae [41,44]. Thus, the concentrations of AgNPs that inhibit spore germination or viability are lower than those causing mycelium growth inhibition. Moreover, the time of exposure to nanoparticles is also important. For example, the effective concentration causing 50% inhibition of mycelial growth (EC_50_) of AgNPs (after 30 h of spores’ exposure) was 1 mg/L for *F. graminearum*, *F. langsethiae*, and *F. poae*, but for *F. proliferatum* and *F. oxysporum*, the EC_50_ was 20.9 and 32.7 mg/L, respectively. In contrast, for spores’ viability, EC_50_ was 1 mg/L for *F. graminearum*, *F. langsethiae*, and *F. poae* but 4.7 and 5.6 mg/L for *F. sporotrichoides* and *F. oxysporum*, respectively [41]. Matras et al. [46] showed that *F. avenaceum* spores’ exposure to chemically synthesized AgNPs (with cysteamine chloride) for 240 h at concentrations of 5–10 mg/L caused the inhibition of further mycelium growth. A shorter time of exposition (24 h) did not show any growth inhibition. However, for *F. equiseti*, the same AgNPs inhibited mycelium growth at concentrations of 2.5 and 10 mg/L after 24 h and 240 h of exposure to spores, respectively. AgNPs prepared using onion, garlic, and ginger extracts by Gutam et al. [43] showed a MIC for *F. avenaceum* of 90–110 mg/L. It was also reported that the linear growth of the mycelium of two *F. avenaceum* isolates was suppressed by AgNPs at 100 mg/L by only 24.5% and 31.2% after 10 days of incubation [40]. The mentioned studies correspond with our results. The MIC value of the bio-AgNPs for spores of both fungi was 125 mg/L (Figure 1), but bio-AgNPs at 200 mg/L inhibited the mycelium growth of *D. pinodes* and *F. avenaceum* by 49% and 55% after 7 days of incubation and by 45% and 26% after 14 days of incubation, respectively (Table 1). Silver nanoparticles used in the present study were previously reported for antibacterial activity (MIC = 1.56 mg/L) by Rilean-Plugaru et al. [55]. Pathogenic bacteria showed greater sensitivity than phytopathogenic fungi in the present study, but in both cases, significant antimicrobial properties of bio-AgNPs were demonstrated, which were further analyzed in in vivo experiments. 

The spread of pea seedlings’ infection was suppressed by bio-AgNP pre-treatment at a similar level to that of fungicides in the case of both phytopathogenic fungi. The level of infection was higher in seedlings infected with *F. avenaceum* than *D. pinodes* (Figure 5C,D), regardless of the same spore concentration being used for inoculation, but the infection symptoms were noticeable in both cases (Figure 3 and Figure 4). *D. pinodes* infection was most effectively reduced by bio-AgNPs at a concentration of 200 mg/L (DI reduction by 94%, fungal gDNA level reduction by 98%), whereas *F. avenaceum* infection was best reduced at 100 mg/L (DI reduction by 86%, fungal gDNA level reduction by 90%), compared to infected seedlings after water pre-treatment. Most studies focus on the in vitro evaluation of the antifungal properties of metal nanoparticles, but in vivo studies under controlled conditions are necessary to verify these findings. AgNPs applied at concentrations of 100 mg/L on plum fruit inhibited the infection of *Botrytis cinerea* by 85% [44], whereas application to black pepper plants before and after inoculation with *Phytophthora capsici* at 10 mg/L reduced the ratio of non-diseased plants to 95.0% and 60.0%, respectively [59]. The infection of tomato seedlings by *F. oxysporum* was also suppressed by AgNPs additives to soil at a concentration of 2000 mg/L [60]. The antifungal activity of bio-AgNPs in planta was confirmed. The present study showed that bio-AgNPs effectively reduced *D. pinodes* and *F. avenaceum* infections in pea seedlings. Furthermore, these nanoparticles did not negatively affect the growth and development of the seedlings (Table 2 and Appendix A). Further metabolomic analyses were performed to reveal if both pathogens, along with nanoparticles, influenced pea seedling polar metabolic profiles in the same way. 

### 3.2. Changes in Polar Metabolic Profiles of Pea Seedlings after Infections

The metabolic profiles of the roots, shoots, and cotyledons of 22-day-old non-inoculated (control) pea seedlings corroborated our previous data presenting the metabolic profiles of 5–7-day-old seedlings [61,62,63], as well as shoots of adult pea plants [64,65]. Glucose and sucrose quantitatively dominated among soluble carbohydrates, homoserine and asparagine were the predominant amino acids, whereas citrate and malate were among the predominant organic acids (Appendix A). The high concentration of the non-proteinogenic amino acid homoserine, an intermediate in the conversion of aspartate into threonine, isoleucine and methionine [66], and asparagine, seems to be a confirmation of the crucial roles of both amino acids in nitrogen metabolism in pea plants. Both are the major forms of nitrogen distributed from source to sink tissues [67], and their concentrations are affected by the drought [64,65] Moreover, homoserine is a component of pea root exudates and present in pea bacteroids, playing the role of the N source, and it can participate in the synthesis of pantothenate in rhizobia [68,69]. 

Plant responses to pathogen infection involve a series of reactions associated with the reinforcement of cell walls, oxidative burst, the activation of the antioxidative system, hormonal adjustment, the synthesis of stress-related biomolecules (e.g., phytoalexins), and proteins related to both secondary [70] and primary metabolism [71], which are common to various fungal infections [72,73,74]. Up to now, the changes in the transcriptome and proteome of pea plants infected with *D. pinodes* or *F. oxysporum* have revealed an increase in the levels of various proteins, i.e., those involved in energy and amino acid metabolism, redox response, the synthesis of pisatin or pathogenesis-related (PR) proteins, and proteins involved in changes in the structures of the cell walls (e.g., lignin biosynthesis, the modification of the degree of methyl esterification of pectins) [2,28,75,76,77,78,79]. However, data on the metabolic responses of pea to fungal infections are scarce. Moreover, no such data about the pea–*F. avenaceum* pathosystem are available. Both *F. oxysporum* and *F. avenaceum* produce mycotoxins such as beauvericin, enniatins, and moniliformin. Additionally, *F. avenaceum* produces fusarins, whereas *F. oxysporum*—fusaric acid and fumonisin, all from the same chemical group—produces polyketides [80]. This might have a similar effect on the metabolic profiles of the pea plants.

Desalegn et al. [81] revealed that the infection of pea (cv. Messire) adult plants (at the 8th to 10th leaf stage) with *D. pinodes* led to changes in the concentrations of citric acid cycle intermediates (↓ oxaloacetate; ↑ citric acid) and amino acids (↓ asparagine; ↑ aspartate, GABA) in leaves, 36 h after inoculation. This was accompanied by increased levels of the crucial enzymes phosphoenolpyruvate carboxylase (PEPC), which synthesizes oxaloacetate from PEP and CO_2_, and NADP-dependent malic enzyme (ME), which catalyzes the oxidative decarboxylation of malic acid and NADP^+^ to produce pyruvic acid, CO_2_, and NADPH. Both products (pyruvate and NADPH) of this reaction are essential in plant–pathogen reactions, participating in the synthesis of flavonoids and lignin [82]. NADPH is also crucial for the reactive oxygen species (ROS) metabolic system (in the ascorbate-glutathione pathway), NADPH-dependent thioredoxin reductase, and apoplastic oxidative burst in most plant–pathogen interactions, and it is a key enzyme in carbon metabolism, playing an important role in plant photosynthesis [82,83]. The negative effect of *D. pinodes* on leaf photosynthesis efficiency was well documented [84] and can be related to the dynamics of plant–pathogen lesions [85,86]. Comparative studies of cultivars more and less resistant to *D. pinodes* infection revealed common metabolomic responses such as increased contents of sugars (but not sucrose), sugar alcohols, and glycolysis/tricarboxylic acid (TCA) cycle intermediates and changes in amino acid content [87]. Moreover, it has been suggested that *D. pinodes* infection in pea plants is regulated by jasmonate (JA) and ethylene (ET) pathways [78,79] and can induce a hypersensitive response that leads to pathogen-induced cell death [28]. 

Although *D. pinodes* and *F. avenaceum* exhibit differences in terms of occurrence, caused symptoms and produced mycotoxins, as well as the infection level in the present study, some similarities in the changes in metabolic profiles after infections were observed. In the shoots of pea seedlings infected after water pre-treatment, the TSCs and TAAs contents were lower than those in the control (non-infected seedlings). In roots, TSCs were depleted as in shoots, but TAA and TOA contents were higher than in the control (Table 3). These changes were caused by the most abundant metabolites—sucrose, homoserine, and asparagine (Appendix A). In roots, glucose and phosphoric acid also differentiated samples’ distribution according to the PCA loading plots (Figure 7E,G). 

In shoots, TAA depletion was related to a dramatic decrease in the asparagine level (approximately 5- and 3-fold under *D. pinodes* and *F. avenaceum* infection, respectively) and a lower concentration of homoserine (more in *F. avenaceum* infection) (Appendix A). However, in roots after *D. pinodes* infection, the asparagine content increased but the homoserine level lowered (Appendix A); in contrast, under *F. avenaceum* infection, no statistical changes were observed in the concentrations of those two amino acids (Appendix A). Amino acid accumulation was also observed in the leaves [88] and roots [89] of soybean after *F. tucumaniae* infection. Asparagine is a transport amino acid involved in N mobilization and translocation, as it can be uploaded directly to the phloem [90]. Homoserine is widely known to be a major amino acid in pea seedlings [63,91]. Moreover, it is considered to be involved in nitrogen transport from storage tissue to growing seedlings [92]. Aspartic acid is a precursor for both homoserine and asparagine [93,94]. Moreover, changes in the concentrations of Hse and Asn are considered to be “host signals”, which can trigger plant responses to infection. Yang et al. [92] reported that homoserine and asparagine trigger the upregulation of pectate lyase D genes (essential for host invasion, expressed only *in planta*) of *Nectria haematococca* during pea seedling infection. Asparagine might also be a source of N for the *Botrytic cinerea* [95] or *F. tucumaniae* [88]. Therefore, the decrease in amino acid content in shoots (especially Hse and Asn) can be related to their mobilization and transport to roots or utilization by pathogens. 

Drastic changes were observed in the case of GABA, which is also worth mentioning. In roots under *D. pinodes* infection, a 10-fold accumulation of GABA in roots (and much lower in shoot and cotyledons) was observed (Figure 8A,C; Appendix A). Under *F. avenaceum* infection, the GABA level duplicated in the roots and increased by approximately ½ in shoots (Figure 8D; Appendix A), remaining at much lower levels than those in seedlings infected with *D. pinodes* (Figure 8). The accumulation of GABA in *Pisum sativum* has been documented earlier in seedlings [96] and older plants under soil drought [64]. GABA is often accumulated under both abiotic and biotic stresses, but the mechanisms of its protective action can vary [97]. Under fungal infection, the accumulation of GABA and GABA shunt (a cytosolic–mitochondrial pathway connecting amino acid metabolism to the TCA cycle via succinate) enables the maintenance of the TCA cycle and regulates C/N metabolism [98]. The role of GABA in response to biotic stresses also includes protection against oxidative and osmotic stress and the regulation of cytosolic pH. GABA can perform nitrogen remobilization as a N storage amino acid, accumulate during stress, and give back N and energy during recovery periods [90]. This, next to sustaining TCA activity, could explain the increased level of GABA (especially in roots) after pathogen infection. 

Another common response of peas to *D. pinodes* and *F. avenaceum* infection was the depletion of proline in shoots and its increased level in roots (Appendix A). Previously, Desalegn et al. [81] and Turetschek et al. [87] reported the accumulation or tendencies to accumulate of proline in the aboveground parts of pea plants under *D. pinodes* infection. Proline is a proteinogenic amino acid that is connected with plant adaptation to various stresses by maintaining osmotic and redox balance, ROS scavenging, and membrane stabilization [99]. Pathogen infection can trigger proline accumulation, which can induce hypersensitive response, whereas proline oxidation can initiate programmed cell death. Moreover, proline catabolism can be controlled by salicylic acid (SA) and JA [71,99]. In our study, distinct necrotic changes were observed in roots after *F. avenaveum* infection (Figure 4), which seems to be consistent with proline accumulation, but in the case of *D. pinodes* infection, we cannot indicate such an association. The lack of such pronounced symptoms in the case of *D. pinodes* infection is probably caused by the lower infection levels. 

Even though TSC concentrations decreased in both roots and shoots, under infection caused by both pathogens, some differences in sugar composition were observed. In shoots, the dramatic depletion of sucrose occurred, but it was not accompanied by adequate changes in glucose and fructose concentrations (Appendix A). In contrast, the sucrose level in the roots remained unchanged, whereas monosaccharide content (except fructose) and *myo*-inositol decreased (Appendix A). Sucrose depletion in pea plants infected with *D. pinodes* is consistent with previous results [87], but glucose and fructose accumulation [81] was not confirmed in our study. However, in the roots of pea infected with *F. oxysporum*, a general decrease in proteins involved in carbohydrate metabolism was observed [76]. Carbohydrates are energetic substrates and carbon nutrients that are essential for providing sufficient energetic resources to maintain plant cells functioning during pathogen infection and activate a plant’s defense mechanisms [100,101]. Sugars and their metabolism contribute to a plant’s immunity as they are associated with increasing cell wall lignification, oxidative burst, the stimulation of flavonoid synthesis (sucrose), and some PR protein expression (by glucose) [71,100]. It has also been suggested that a lowered hexose/sucrose ratio during infection can trigger PR gene expression [102]. Infection forces plants to undergo carbon nutrient mobilization, accompanied by photosynthesis reduction, increased respiration, and alterations in nitrogen and lipid metabolism. Such a shift from source to sink metabolic status induces the long-distance transport of carbohydrates in the form of sucrose from healthy to infected parts of the plant [71,101]. Moreover, pathogens can be an additional “sink” of carbon and nitrogen compounds and even modulate the metabolism of plants to satisfy their own nutritional needs [100,101]. Excessive energy requirements during pathogen infection seem to explain the observed decrease in soluble carbohydrate content in peas infected with *D. pinodes* and *F. avenaceum*. 

In our study, only slight changes in the levels of citrate and malate were found in infected pea seedlings (Appendix A), especially in the roots of seedlings infected with *F. avenaceum* (↑ citrate, malate, succinate), but minor changes were also noticed in seedlings after *D. pinodes* infection (↓ citrate, ↑ succinate in roots; ↑ malate, succinate in shoots). Similar changes were reported by Desalegn et al. [81], where metabolites such as citric acid, succinic acid and oxaloacetate exhibited lower levels in plants infected with *D. pinodes* than in healthy ones, except for malate, whose level was similar. Such subtle changes suggest only minor distribution in the TCA cycle. All of the above-mentioned similarities between the effects of pathogens on the polar metabolic profiles of pea seedlings may be indicated by the fact that *D. pinodes* and *F. avenaceum* are hemibiotrophs [103,104]—after the initial biotrophic phase, they turn to necrotrophy, triggering similar defense mechanisms. 

### 3.3. Effect of Bio-AgNP Pre-Treatment on Changes in Polar Metabolic Profiles of Pea Seedlings after Infections

The effect of bio-AgNPs on changes in the metabolome of roots and shoots was somewhere between the effects caused by the fungicide and the control seedlings. However, some differences were observed between pre-treatment with nanoparticles in cases of infection with *D. pinodes* and *F. avenaceum*. Inconsistent metabolic responses may depend on the pathogen species or the level of infection. In the case of seedlings after pre-treatment with bio-AgNPs and infected with *F. avenaceum*, the TIPM content was elevated, and soluble sugars were at a similar level to the control (Table 3). Also, Asn and GABA (only in shoots) concentrations were increased (Appendix A). In contrast, TIPMs, in the case of seedlings after pre-treatment with bio-AgNPs and infected with *D. pinodes*, were decreased compared to control seedlings (Table 3). In roots, Asn and GABA contents were at similar level as in the control, whereas their elevated level was observed under infection in water pre-treatment seedlings (Appendix A). An interesting common response between infections was observed in roots—proline and serine concentrations were at a similar level as those in the control seedlings (Appendix A). These results suggest that after nanoparticle pre-treatment, infected plants maintain their functioning at a level similar to that of non-infected plants or mobilize their metabolism. After short-time immersion in bio-AgNPs, nanoparticles could remain on the surface of pea seedlings or enter the plant through the stomata, rhizodermis, root tip, and hairs. Absorption through the epidermis is also possible but is limited due to the limitation of pores’ sizes [105]. Therefore, two possible mechanisms may underlie the suppression of pea seedling infection. This can be caused by the direct interaction of nanoparticles with fungal cells or by enhancing the plant’s immune response. The antifungal properties of metal nanoparticles are primarily associated with their ability to disrupt fungal cell integrity and functioning by affecting membrane permeability, inducing the generation of ROS, and causing DNA and protein damage [39]. The mentioned mechanisms of direct action are considered to be the most important and widely analyzed in the context of nanoparticle–pathogen interactions. It is not clear whether metal nanoparticles activate plants’ defense mechanisms [38]. However, in the context of the nanoparticle–pathogen–plant relationship, nanoparticles are increasingly being considered as factors supporting plant resistance to biotic stresses [73], as they have also been reported to alleviate abiotic stresses [106,107]. 

Nanoparticles pose as resistance-inducing compounds that can trigger systemic acquired resistance (SAR) [108]. This type of induced resistance is caused by local contact with a pathogen or a compound/elicitor-inducing resistance reaction. Resistance is induced locally and then spreads to the entire plant via signal transduction pathways. The salicylic acid (SA) pathway is most frequently associated with SAR and is responsible for the activation of pathogenesis-related (*PR*) genes [109]. Decreased disease severity, along with the upregulation of *PR* genes, was observed in *Arabidopsis thaliana* and *Nicotiana benthamiana* treated with a nanocomplex of silver and silica, SiO_2_NPs and AgNPs, after infection with the bacterial pathogens *Pseudomonas syringae* pv. *tomato* [110], *P. syringae* [109], and *P. syringae* pv. *tabaci* [111], respectively, in addition to decreased disease severity. In tobacco plants treated with AgNPs and infected with *P. syringae* pv. *tabaci*, increased activity of peroxidase (POD) and polyphenol oxidase (PPO) was observed, as well as increased production of ROS [111]. However, AgNPs used against *Clavibacter michiganensis* subsp. *michiganensis* did not upregulate *PR* genes in tomato plants but significantly suppressed disease severity. This suggests that nanoparticles can also mediate other induced resistance mechanisms [112]. Therefore, as the antifungal properties of bio-AgNPs have been confirmed, further research is necessary to clarify the role of nanoparticles in pea protection against *D. pinodes* and *F. avenaceum*. 

## 4. Materials and Methods

### 4.1. Bio-Synthesized Silver Nanoparticles and Their Characterization

The silver nanoparticles used in this study were previously synthesized and fully described by Railean-Plugaru et al. [55]. These nanoparticles, specifically nanocomposites, were synthesized using a *Lactobacillus paracasei* post-culture medium and found to have a complex structure of a metallic silver core and organic coat of a layer consisting of the metabolites secreted naturally by the bacteria. Their size ranged between 5 and 30 nm [113], and the mean size was 18 ± 2.4 nm [55]. The preparation of the nanoparticles was adapted for the experiments. 

In this study, the elemental composition of the bio-synthesized silver nanoparticles (bioAgNPs) was determined using a Scanning Electron Microscope (SEM, LEO 1430 VP, Leo Electron Microscopy Ltd., Cambridge, United Kingdom) equipped with an Energy Dispersive X-ray (EDX) detector (XFlash 4010, Bruker AXS, Bremen, Germany). Samples of the silver powder were affixed to a microscope holder using a conducting carbon strip to ensure stability during analysis.

The core size and morphology of the bio-AgNPs were characterized using Transmission Electron Microscopy (TEM, FEI Tecnai F20 X-Twintool, FEI Europe, Frankfurt/Main, Germany). A droplet of the nanoparticle suspension was placed onto carbon-coated copper grids and left to dry, allowing the solvent to evaporate completely before imaging.

Interactions between the metallic surfaces and organic ligands were investigated using an AV G Microtech X-ray Photoelectron Spectrometer (XPS). The surface composition of the nanoparticle pellets was examined under a high vacuum condition of (10^−9^) mbar using MgKα non-monochromatized X-ray radiation (hν = 1253.6 eV) to enhance the Ag3d signal detection. Curve fitting and data analysis were performed with CasaXPS software (version 2.3), utilizing a Shirley baseline for background subtraction and employing a mix of Lorentzian and Gaussian lines for signal decomposition. Asymmetric functions were used for simulating metallic states, and multiplet structures were based on the models detailed in Biesinger et al. [114]. 

The content of elemental silver in the bio-AgNPs was quantified using Inductively Coupled Plasma Optical Emission Spectrometry (ICP-OES, AVIO 220, PerkinElmer, Warszawa, Poland). Approximately 1 mg of bio-AgNPs was dissolved in 1 mL of concentrated nitric acid (65% HNO_3_), diluted with doubly distilled water (ddH_2_O) to achieve a 5% (*v*/*v*) HNO_3_ solution for analysis.

### 4.2. Fungal Cultures and Spore Preparation

In these experiments, we used *Didymella pinodes* strain no. CBS 107.45, purchased from the Westerdijk Fungal Biodiversity Institute (Utrecht, The Netherlands), and *Fusarium avenaceum* strain no. A232/2019, acquired from the collection of the Department of Entomology, Phytopathology, and Molecular Diagnostics, the University of Warmia and Mazury (Poland). 

Cultures of *D. pinodes* and *F. avenaceum* were carried out in petri dishes on a potato-dextrose agar (PDA; potato extract 20%, glucose 1.6%, agar 1.8%) at 22 °C (day/night 12 h/12 h, in a climatic chamber). Sterile water was applied to the surface of a petri dish with the grown 21-week-old culture of *D. pinodes* and *F. avenaceum*, and spores were suspended with a sterile glass spreader. The suspension was transferred to a 25 mL glass bottle and use immediately in a resazurin assay or shaken for 24 h at 22 °C and 110 rpm and used for plant infection. The spore suspension was filtered through sterile gauze to remove the residual of mycelium and then diluted to a proper density of spores (colony forming unit; CFU) per mL, which was determined using a Brücker hemocytometer. For the MIC value the determination of 2 × 10^6^ CFU/mL was used, whereas for seedlings, infection with 1 × 10^7^ CFU/mL was used. 

### 4.3. MIC Value Determination—Resazurin Assay

To determine the minimum inhibitory concentration (MIC) for fungal spores, the resazurin assay was performed, according to Saker et al. [115] with modifications. Resazurin solution was prepared by dissolving 10 mg of resazurin sodium salt in 3 mL of sterile water. The bio-AgNPs at concentrations of 1200 mg/L were suspended in double-distilled sterile water via sonication (Sonic-3, 310 W, 40 KHz, POLSONIC, Pałczyński, Poland) for proper nanoparticle distribution (2 times for 30 min), and then used for serial dilution. The assay involved preparing 2-fold dilutions of bio-AgNPs (stock solution—1200 mg/L) in a 96-well microtitration plate with concentrations as follows: 500, 250, 125, 62.5, 31.25, and 15.625 mg/L (final concentrations in wells). To each well with potato-dextrose broth (PDB; potato extract 20%, glucose 1.6%) and bio-AgNPs (v = 100 µL), 10 µL of prepared spores or sterile water and 10 µL of resazurin solution were added (total well volume = 120 µL). Two types of controls were used: positive control—wells only with PDB, spores, and resazurin solution; negative control—wells with diluted bio-AgNPs or PDB with resazurin solution without spores. The plates were incubated for 4 days at 22 °C (day/night, 12 h/12 h). The lowest concentration of nanoparticles for which no discoloration occurred was considered to be the MIC.

### 4.4. Inhibition of Mycelial Fungal Growth—Poisoned Food Technique

Bio-AgNPs were added to sterile, unsolidified PDA (cooled to 60 °C) at final concentrations of 10, 100, 150, and 200 mg/L and suspended via sonication 2 times for 30 min (temp. 60 °C) for proper nanoparticle distribution. PDA, with the addition of the fungicide, was used as a positive control for the comparison of the fungistatic effect of bio-AgNPs with commercially available products. The fungicide for *D. pinodes* was Amistar 250 SC (Syngenta Poland, Warsaw, Poland) with 250 g/L (22.81%) of azoxystrobin, and that for *F. avenaceum* was Toledo Extra 430 SC (Rotam Agrochemical Europe, Lyon, France) with 430 g/L (33.29%) of tebuconazole. According to the manufacturers’ recommendations, Amistar and Toledo were used at concentrations of 1.143 mL/L and 1.5 mL/L, respectively.

The petri dishes were inoculated at the center with 6 mm agar plugs from the monospore cultures of *F. avenaceum* (6 plates per variant, in three series) and incubated for 14 days at 22 °C, (day/night 12 h/12 h). The mycelial diameter was measured after 7 and 14 days. Mycelial growth inhibition (MGI) was calculated based on Equation (1), according to Balouiri et al. [116], with minor modifications: MGI (%) = ((Dc − d) − (Ds − d))/(Dc − d) × 100%(1)MGI—mycelial growth inhibition; Dc—the diameter of mycelium growth in the control plate; Ds—the diameter of mycelium growth in the plate with nanoparticles; d—the diameter of the mycelium agar plug used for inoculation.

### 4.5. Plant Material 

The pea (*Pisum sativum* L.) seed cultivar Nemo, purchased from Danko Hodowla Roślin (Choryń, Poland), were germinated for 4 days in rolls of wet filter paper placed in 250 mL glass cylinders (22 °C, in the dark, in a climatic chamber Snijders Scientific, Tilburg, The Netherlands). They were then transferred to 10 mL probes with distilled water so that the roots were immersed in water and incubated at 22 °C (day/night, 12 h/12 h; climatic chamber Snijders Scientific, The Netherlands) for the next three days. Water in the probes with seedlings was replenished daily. The 7-day-old healthy seedlings with properly developed epicotyl and primary root were used for the infection experiments. 

### 4.6. Plant Infection 

Plant infection experiments were independently performed for each pathogen. Seven-day-old pea seedlings were divided into 5 groups with 3 replicates of 16 seedlings each. Groups 1 and 2 were immersed for 5 s in water, group 3 was immersed in a fungicide, and groups 4 and 5 were immersed in bio-AgNPs (at concentrations of 100 and 200 mg/L, respectively). The bio-AgNPs at concentrations of 100 and 200 mg/L were suspended in double-distilled water via sonication (2 times for 30 min), and then Tween80 was added (concentration of 0.1% in the suspension) to improve nanoparticle adhesion (sonicated for 15 min). The fungicides used for *D. pinodes* and *F. avenaceum* infection experiments were Amistar 250 SC (azoxystrobin) and Toledo Extra 430 SC (tebuconazole), respectively, as in the in vitro experiment (Section 4.4.). After 24 h, seedlings were inoculated with spores of *D. pinodes* or *F. avenaceum* prepared as described in Section 4.2. Tween80 was added to the spore suspension to improve adhesion at a final concentration of 0.1% in the suspension. According to the infection mechanism of *D. pinodes* and *F. avenaceum*, seedlings infected with *D. pinodes* were inoculated on the shoot, whereas those infected with *F. avenaceum* were inoculated on the base of the shoot and root. The seedlings were incubated in a climatic chamber for 14 days (temp. 22 °C, day/night 12 h/12 h), and water was replenished daily.

The morphology of the plants (fresh and dry weight, and the length of shoots and primary roots) was measured, and the disease index was determined 14 days post-infection for all seedlings from each treatment (in 22-day-old seedlings). Half of the pea seedlings from each replication of each treatment were divided into shoots, roots, and cotyledons; frozen in liquid nitrogen; and stored at −80 °C for further metabolomic analyses. The other half of the seedlings was also collected, frozen in liquid nitrogen, and stored at −80 °C for further qPCR analysis. 

### 4.7. Disease Index

The assessment of the health of the pea seedlings (all seedlings from each treatment) was performed according to the modified scale of Hillstrand and Auld [117], in which successive degrees of infestation corresponded to the percentage of disease symptoms present in the plant. The disease index of pea was calculated according to the McKinney Formula (2) [118]: DI (%) = (Σ (a × b) × 100%)/(N × I)(2)DI—the disease index; Σ (a × b)—the sum of the products after multiplying the number of plant organs examined by the given degree of infestation; N—the total number of organs examined; I—the highest degree of infestation on the scale.

### 4.8. qPCR Analysis

qPCR was performed to quantify the level of infection expressed in the presence of fungal DNA. The collected tissues (whole seedlings) were homogenized in liquid nitrogen using a mortar and pestle. The isolation of gDNA was performed gDNA-isolation for Maxwell^®^ 16 FFS (Promega, Madison, WI, USA) with minor modifications. In brief, 600 µL of CTAB buffer, 10 µL of Proteinase K, and 4 µL of RNase A solution were added to 200 mg of homogenized material. The samples were vortexed and incubated at 65 °C for 30 min with continuous shaking (500 rpm). After centrifugation (14,000× *g* at 4 °C for 10 min), 300 µL of the supernatant and 300 µL of Lysis Buffer were added to the Maxwell instrument. 

The quality and quantity of gDNA were measured using a NanoDrop ND 2000c spectrophotometer (Thermo Scientific, Waltham, MA, USA). Matrices with high-quality parameters were used in further studies. The qPCR quantification of the level of infection expressed by the presence of fungal DNA was performed using primers specific for *D. pinodes* [13] and *F. aveneceum* [119] (Table 4). qPCR was performed using an ABI Prism 7500 Fast instrument (Applied Biosystems, Waltham, MA, USA). The reaction mixture contained 12.5 µL of TaqMan Fast Universal PCR Master Mix (Applied Biosystem, ThermoFisher Scientific, Waltham, MA, USA), 10 pM probes labeled at the ends with 5′- FAM and 3′- MGB as a quencher, 10 pM primers, 4.5 µL of deionized water, 5 µL of gDNA, and a total volume of 25 μL. The amplification conditions were the same as those described by Okorski et al. [120]. qPCR was performed in duplicates for each test variant. The negative control was deionized, sterile water, while the positive control was gDNA extracted from *D. pinodes* and *F. avenaceum* 21-day-old cultures. Quantitative calculations of qPCR were performed according to the method described by Livak and Schmittgen [121].

### 4.9. Polar Metabolite Analyses 

#### 4.9.1. Extraction of Polar Metabolites 

Tissue samples of infected seedlings were lyophilized and pulverized in a mixer mill (MM200, Retsch, Haan, Germany). Extraction was performed according to Fiehn [122], with modifications according to Szablińska-Piernik and Lahuta [64]. The extraction of 40–42 mg of milled tissue (at least 3 biological replicates) was performed with 900 μL of 50% methanol at 70 °C for 30 min with continuous shaking (500 rpm; Thermo-shaker MS-100 ALLSHENG, Hangzhou, China). Ribitol was used as the internal standard (100 μL of 1 mg/mL ribitol added to the extraction mixture). Subsequently, the homogenates were cooled and centrifuged (20,000× *g* at 4 °C for 20 min). The supernatants were mixed with cold chloroform to remove non-polar compounds. The samples were then dried in chromatographic vials and stored in desiccators until chromatographic analysis.

#### 4.9.2. GC-FID and GC-MS Analyses

Tissues metabolic profiling was performed using a gas chromatograph GC2010 Nexia (Shimadzu, Kyoto, Japan) with a flame ionization detector (FID) for robust quantitative analyses of metabolites and a gas chromatograph coupled with mass spectrometry (QP-GC-2010, Shimadzu, Japan) to confirm accurate metabolite identification, according to the protocols described by Szablińska-Piernik and Lahuta [64]. The dried samples were derivatized in two steps: O-methoxamine hydrochloride and a mixture of N-methyl-N-trimethylsilyl-trifluoroacetamide (MSTFA) with pyridine (1:1, *v*/*v*). The mixtures of trimethylsilyl (TMS)-derivatives were separated on a ZEBRON ZB-5MSi Guardian capillary column (Phenomenex, Torrance, CA, USA). Metabolites were identified and characterized via the comparison of their retention times (RT), retention indices (RI, determined according to the saturated hydrocarbons), and mass spectra of original standards derived from Sigma-Aldrich (Sigma-Aldrich, Merck, Burlington, MA, USA) and the NIST library (National Institute of Standards and Technology).

### 4.10. Statistical Analyses 

The results are the mean of 3 independent replicates, and they were subjected to one-way ANOVA with a post hoc test (Tukey, if overall *p* < 0.05) using Statistica software (version 12.0; StatSoft, Tulsa, OK, USA). Additionally, Pearson’s correlation (*p* < 0.001) was performed using Statistica software (version 12.0; StatSoft, Tulsa, OK, USA) to measure the linear correlation between the disease index and the level of fungal gDNA in plant tissues 14 days after inoculation. Graphs were prepared using GraphPad Prism, version 8 (GraphPad Software, San Diego, CA, USA). Multivariate statistics of metabolomic data were analyzed using principal component analysis (PCA) and performed using COVAIN [123], a MATLAB toolbox including a graphical user interface (MATLAB version 2013a; Math Works, Natick, MA, USA).

## 5. Conclusions

In the present study, we confirmed the antifungal properties of bio-AgNPs against *D. pinodes* and *F. avenaceum*, both in vitro and in vivo. Most studies focus on the early response of adult pea plants to pathogens and the subsequent mobilization of plants’ metabolism to counteract the attack. Our study demonstrated a rather long-term, established response/interaction of the pathogen to/with the host. The main changes concerned amino acid and soluble carbohydrate concentrations, which were similar in the case of both pathogens. The depletion of amino acid content in shoots (especially Hse and Asn) is probably associated with mobilization, transport to the roots, or utilization by pathogens. Similarly, a decrease in soluble sugars can be associated with the increased energy demand of the plant, which, during infection, is often not compensated via photosynthesis, but sugars can also be utilized by fungi during the infection process. 

Bio-AgNP pre-treatment of seedlings infected with *F. avenaceum* could mobilize the plant to respond to the pathogen as the TIPM content was elevated. In the case of seedlings infected with *D. pinodes* after pre-treatment with bio-AgNPs, nanoparticles were rather involved in maintaining the plants’ functioning, which was observed to have decreased TIPM levels compared to control seedlings and Asn and GABA contents similar to that of the control, elevated under infection in water pre-treatment seedlings. Bio-AgNP pre-treatment in both infections contributed to the depletion of proline and serine concentrations in roots compared to water pre-treatment, maintaining a level similar to those of control seedlings. Apart from the direct effect on fungi, the nanoparticles also suppressed the spread of *D. pinodes* and *F. avenaceum* infections and changed the polar metabolic profiles of pea seedlings during infections. The possible dual effect of nanoparticles, as protectors and defense response inductors, should be considered in further in vivo studies. The presented results suggest that bio-AgNPs could be used in pea plant protection against selected pathogens, with an effectiveness equal to those of currently available fungicides.

## Figures and Tables

**Figure 1 ijms-25-04525-f001:**
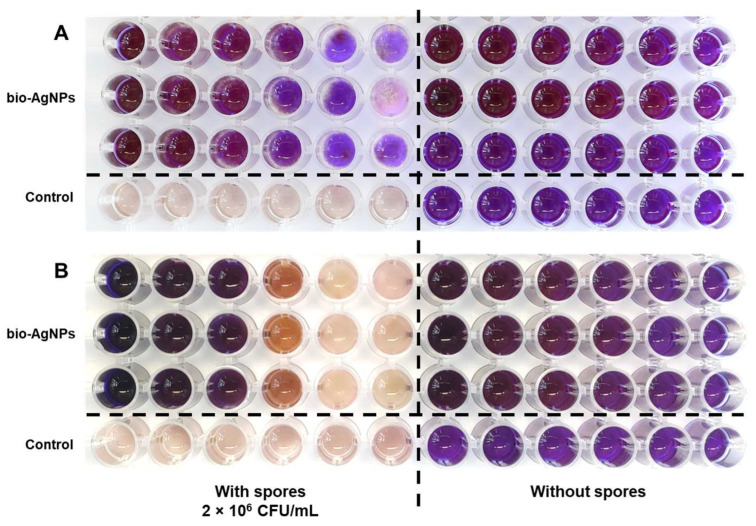
The MIC values of bio-AgNPs against *D. pinodes* (**A**) and *F. avenaceum* (**B**), as determined via a resazurin assay with serial dilutions of silver nanoparticles. Bio-AgNPs were used at concentrations of 500, 250, 125, 62.5, 31.25, and 15.625 mg/L (from left to right); control—potato-dextrose broth (PDB) without nanoparticles.

**Figure 2 ijms-25-04525-f002:**
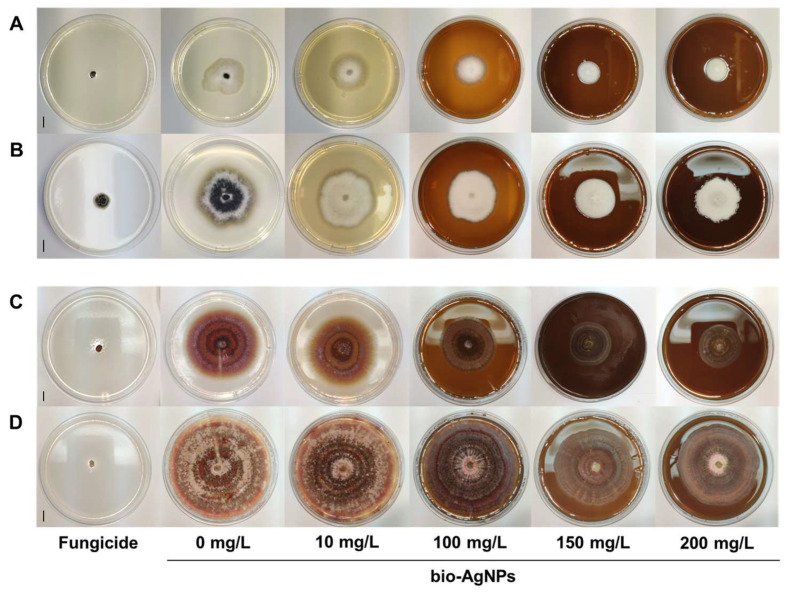
Inhibition of mycelial fungal growth after 7 and 14 days of incubation of *D. pinodes* ((**A**) and (**B**), respectively) and *F. avenaceum* ((**C**) and (**D**), respectively) on a potato-dextrose agar (PDA) with different concentrations of bio-AgNPs. For comparison, the fungicides Amistar 250 SC (22.82% azoxystrobin) and Toledo Extra 430 SC (33.29% tebuconazole) were used for *D. pinodes* and *F. avenaceum*, respectively. The vertical scale bars equal 10 mm.

**Figure 3 ijms-25-04525-f003:**
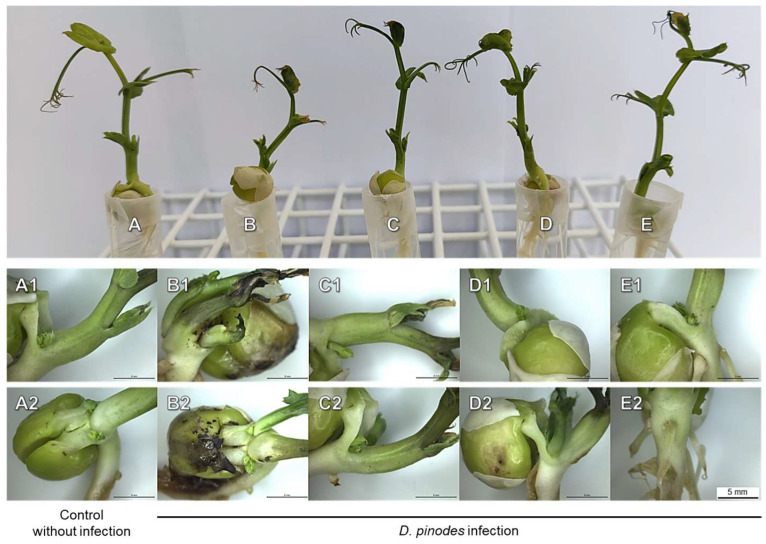
The 22-day-old pea seedlings not infected (control, (**A**)) and 14 days after inoculation with *D. pinodes* after the seedlings’ short-term immersion in water (**B**), a fungicide (Amistar 250 SC with 22.82% azoxystrobin), (**C**) or bio-AgNPs ((**D**,**E**), at concentrations of 100 and 200 mg/L, respectively). The symptoms of infection were enlarged in the pictures in the two bottom rows: control (**A1**,**A2**); seedlings treated with water (**B1**,**B2**), fungicide (**C1**,**C2**) and bio-AgNPs at concentration of 100 mg/L (**D1**,**D2**) and 200 mg/L (**E1**,**E2**).

**Figure 4 ijms-25-04525-f004:**
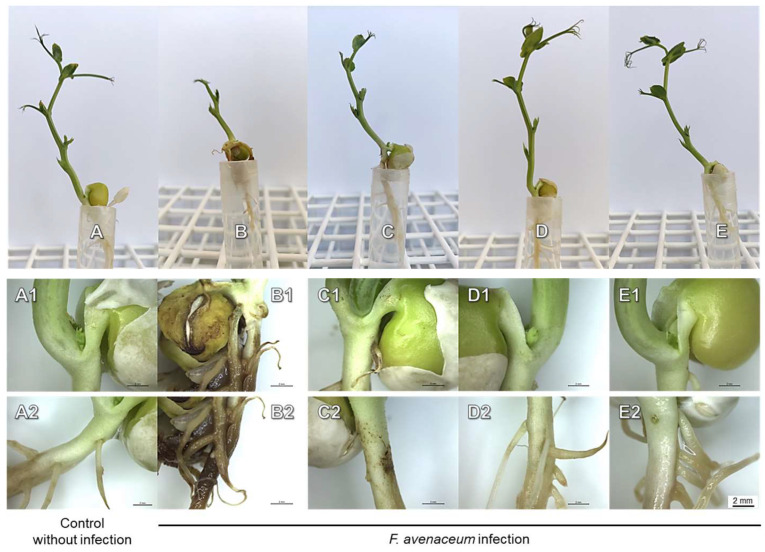
The 22-day-old pea seedlings not infected (control, (**A**)) and 14 days after inoculation with *F. avenaceum* after the seedlings’ short-term immersion in water (**B**), a fungicide (Toledo Extra 430 SC with 33.29% tebuconazole), (**C**) or bio-AgNPs ((**D**,**E**), at concentrations of 100 and 200 mg/L, respectively). The symptoms of infection were enlarged in the pictures in the two bottom rows: control (**A1**,**A2**); seedlings treated with water (**B1**,**B2**), fungicide (**C1**,**C2**) and bio-AgNPs at concentration of 100 mg/L (**D1**,**D2**) and 200 mg/L (**E1**,**E2**).

**Figure 5 ijms-25-04525-f005:**
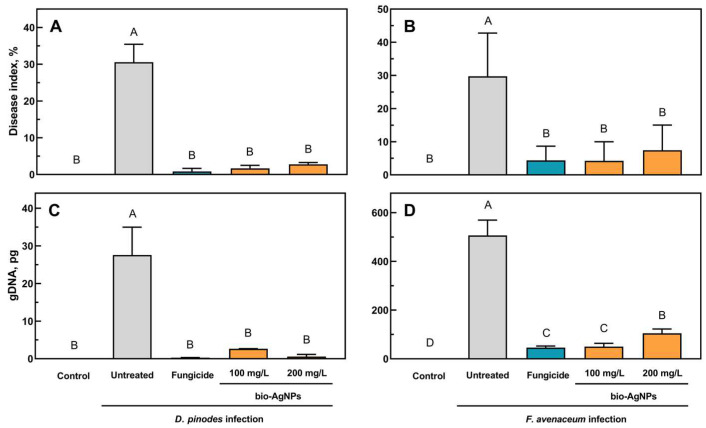
The disease index (the percentage of disease symptoms) of pea seedlings treated with water, a fungicide, and bio-AgNPs, caused by *D. pinodes* (**A**) and *F. avenaceum* (**B**), and the quantity of *D. pinodes* (**C**) and *F. avenaceum* (**D**) in the pea seedlings 14 days post-inoculation. Values are the means of 3 replicates + sd. The same letters above the bars indicate statistically insignificant (*p* < 0.01) differences based on ANOVA and Tukey’s post hoc corrections.

**Figure 6 ijms-25-04525-f006:**
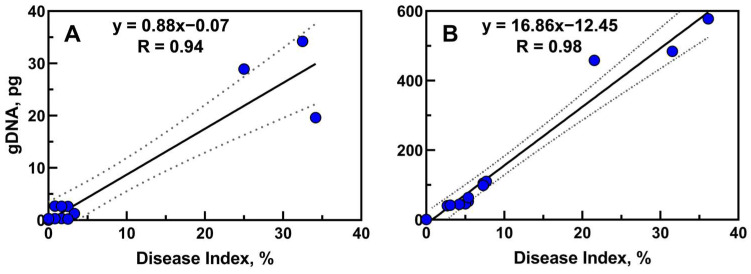
Linear regression (*p* < 0.001) between fungal gDNA and the disease index in seedlings infected with *D. pinodes* (**A**) and *F. avenaceum* (**B**). R—Pearson’s correlation coefficients (*p* < 0.001).

**Figure 7 ijms-25-04525-f007:**
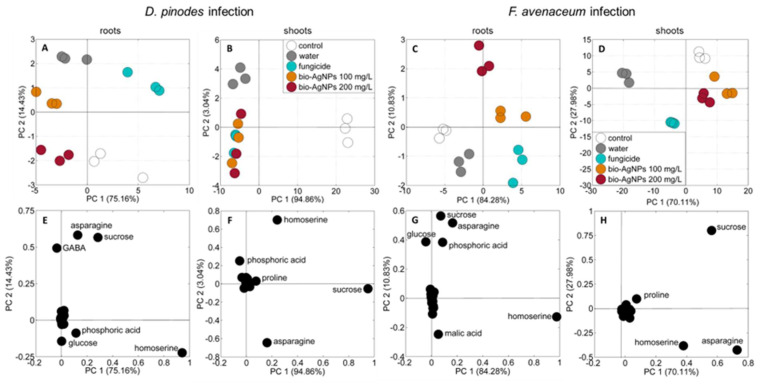
PCA score (**A**–**D**) and loading plots (**E**–**H**) of the metabolic profiles of the roots and shoots of 22-day-old seedlings of pea (*Pisum sativum* L.), 14 days after *D. pinodes* (**A**,**B**,**E**,**F**) or *F. avenaceum* (**C**,**D**,**G**,**H**) inoculation. Abbreviations: control—non-infected seedlings; water, fungicide, bio-AgNPs 100 and 200 mg/L—seedlings pre-treated with water, a fungicide, or bio-AgNPs (at 100 and 200 mg/L), respectively before infection with *D. pinodes*.

**Figure 8 ijms-25-04525-f008:**
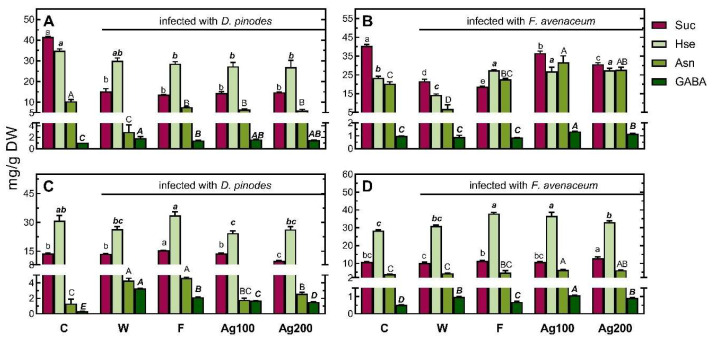
The concentrations of sucrose (Suc), homoserine (Hse), asparagine (Asn), and γ-aminobutyric acid (GABA) in the shoots (**A**,**B**) and roots (**C**,**D**) of 22-day-old seedlings of pea (*Pisum sativum* L.), 14 days after *D. pinodes* (**A**,**C**) or *F. avenaceum* (**B**,**D**) inoculation. Values (in mg/g DW) are the means of 3 replicates + SD. The same letters (a–e; ***a***–***c***; A–D; ***A***–***E***; separately for sucrose, Hse, Asn and GABA, respectively) above the bars indicate statistically insignificant (*p* < 0.05) differences based on ANOVA and Tukey’s post hoc corrections. Abbreviations: *C*—control, not infected seedlings; *W*, *F*, *Ag100*, and *Ag200*—seedlings infected after short-term immersion in water, a fungicide (Amistar 250 SC against *D. pinodes* and Toledo Extra 430 SC against *F. avenaceum*), or bio-AgNPs at 100 and 200 mg/L, respectively.

**Table 1 ijms-25-04525-t001:** Mycelium diameter and mycelial growth inhibition (MGI) after 7 and 14 days of *D. pinodes* and *F. avenaceum* incubation on PDA without fungicide/bio-AgNPs and with different concentrations of bio-AgNPs, compared with the effect of commercially available fungicides (Amistar 250 SC with 22.82% azoxystrobin and Toledo Extra 430 SC with 33.29% tebuconazole for *D. pinodes* and *F. avenaceum*, respectively).

			After 7 Days		After 14 Days	
			Mycelium Diameter (mm)	MGI (%)	Mycelium Diameter (mm)	MGI (%)
*D. pinodes*	Control		41.5 ^A^	-	73.2 ^A^	-
	Fungicide		6.0 ^D^	100.00 ^A^	12.0 ^E^	91.06 ^A^
	bio-AgNPs	10 mg/L	41.2 ^AB^	0.10 ^B^	64.7 ^B^	12.82 ^C^
		100 mg/L	33.6 ^AB^	21.81 ^B^	49.4 ^C^	22.73 ^BC^
		150 mg/L	29.2 ^B^	33.68 ^B^	49.4 ^C^	21.95 ^BC^
		200 mg/L	23.8 ^C^	49.23 ^AB^	38.5 ^D^	45.06 ^B^
*F. avenaceum*	Control		62.6 ^A^	-	78.9 ^A^	-
	Fungicide		6.0 ^C^	100.00 ^A^	6.0 ^C^	100.00 ^A^
	bio-AgNPs	10 mg/L	63.7 ^A^	−2.24 ^C^	85.0 ^A^	−8.96 ^C^
		100 mg/L	37.4 ^B^	44.89 ^B^	69.3 ^B^	13.30 ^B^
		150 mg/L	31.3 ^B^	55.31 ^B^	64.1 ^B^	19.99 ^B^
		200 mg/L	31.3 ^B^	55.43 ^B^	59.8 ^B^	25.92 ^B^

The same letters beside the values indicate statistically insignificant (*p* < 0.05) differences (valid separately for each fungus and each parameter after 7 and 14 days) based on ANOVA and Tukey’s post hoc corrections.

**Table 2 ijms-25-04525-t002:** The fresh weight (FW) and lengths of the shoot and root of 22-day-old pea seedlings developed without infection (control) or infected (on the 8th day of germination, DG) with *D. pinodes* or *F. avenaceum* after pre-treatment with water, a fungicide, or bio-AgNPs (at 100 and 200 mg/L).

			Control	Seedlings Infected after Short-Term Immersion in			
				Water	Fungicide *	Bio-AgNPs	
						100 mg/L	200 mg/L
*D. pinodes* infection	FW (mg)	shoots	211.1 ^a^	136.5 ^b^	160.1 ^ab^	171.3 ^ab^	198.0 ^ab^
		roots	195.0 ^ab^	159.3 ^b^	205.5 ^ab^	210.2 ^ab^	218.6 ^a^
	Length (mm)	shoots	33.4 ^a^	31.7 ^a^	30.3 ^a^	36.3 ^a^	38.5 ^a^
		roots	54.7 ^bc^	47.7 ^c^	55.0 ^b^	59.9 ^ab^	62.2 ^a^
*F. avenaceum* infection	FW (mg)	shoots	307.2 ^ab^	212.7 ^b^	286.0 ^ab^	331.4 ^a^	307.4 ^a^
		roots	363.6 ^a^	364.9 ^a^	333.0 ^a^	354.2 ^a^	290.3 ^a^
	Length (mm)	shoots	56.7 ^ab^	44.0 ^b^	57.4 ^ab^	60.6 ^a^	64.4 ^a^
		roots	97.6 ^a^	95.3 ^a^	97.4 ^a^	99.0 ^a^	96.7 ^a^

Means of 3 replicates. The same letters (a–c) indicate statistically insignificant (*p* < 0.05) differences (valid separately for roots and shoots) based on ANOVA and Tukey’s post hoc corrections. * Amistar 250 SC against *D. pinodes* and Toledo Extra 430 SC against *F. avenaceum* were used. No significant changes were found in shoots’ length (Table 2), cotyledons’ FWs, and the DWs of roots, shoots, and cotyledons (Appendix A).

**Table 3 ijms-25-04525-t003:** The concentration of total identified polar metabolites (TIPMs), including total soluble carbohydrates (TSCs), total amino acids (TAAs), total organic acids (TOAs), and total remaining compounds (TRCs), in the roots and shoots of 22-day-old pea seedlings (*Pisum sativum* L.), without infection (control) and 14 days post-inoculation with *D. pinodes* or *F. avenaceum* (on the 8th day of germination, DG). Before inoculation, the seedlings (at 7th DG) were short-term immersed in water, a fungicide, and a suspension of bio-AgNPs (at 100 and 200 mg/L).

			Control	Seedlings Infected after Short-Term Immersion in			
				Water	Fungicide *	Bio-AgNPs	
						100 mg/L	200 mg/L
		Metabolites	mg/g DW				
*D. pinodes* infection	Roots	TIPMs, including the following:	64.90 ^b^	65.53 ^b^	74.17 ^a^	58.12 ^c^	56.21 ^c^
		TSCs	17.16 ^ab^	15.74 ^c^	17.70 ^a^	16.83 ^b^	12.29 ^d^
		TAAs	36.19 ^bc^	39.06 ^b^	45.25 ^a^	31.33 ^c^	34.11 ^bc^
		TOAs	3.57 ^bc^	3.74 ^a^	3.68 ^ab^	3.44 ^c^	3.24 ^d^
		TRCs	7.98 ^a^	6.99 ^c^	7.55 ^b^	6.53 ^d^	6.57 ^d^
	Shoots	TIPMs, including the following:	121.26 ^a^	82.87 ^b^	80.22 ^b^	80.25 ^b^	80.84 ^b^
		TSCs	47.41 ^a^	20.46 ^b^	18.00 ^c^	18.77 ^c^	19.01 ^bc^
		TAAs	59.63 ^a^	44.82 ^b^	46.27 ^b^	45.21 ^b^	45.48 ^b^
		TOAs	3.54 ^d^	4.32 ^b^	3.91 ^c^	3.94 ^c^	5.06 ^a^
		TRCs	10.68 ^d^	13.25 ^a^	12.03 ^b^	12.34 ^b^	11.29 ^c^
*F. avenaceum* infection	Roots	TIPMs, including the following:	59.52 ^c^	64.05 ^b^	72.73 ^a^	73.31 ^a^	71.45 ^a^
		TSCs	14.65 ^b^	12.84 ^c^	13.63 ^bc^	14.80 ^b^	17.11 ^a^
		TAAs	36.28 ^c^	41.36 ^b^	48.75 ^a^	48.21 ^a^	43.96 ^b^
		TOAs	2.86 ^c^	4.08 ^a^	4.20 ^a^	3.40 ^b^	3.03 ^c^
		TRCs	5.72 ^c^	5.78 ^c^	6.15 ^c^	6.90 ^b^	7.34 ^a^
	Shoots	TIPMs, including the following:	117.80 ^b^	72.23 ^d^	103.45 ^c^	132.60 ^a^	119.60 ^b^
		TSCs	47.72 ^a^	29.64 ^c^	27.86 ^c^	46.12 ^a^	38.52 ^b^
		TAAs	58.82 ^c^	30.31 ^d^	61.82 ^bc^	73.94 ^a^	68.74 ^ab^
		TOAs	3.12 ^b^	3.08 ^b^	3.68 ^a^	3.12 ^b^	3.25 ^b^
		TRCs	8.13 ^c^	9.20 ^b^	10.10 ^a^	9.42 ^b^	9.10 ^b^

* Amistar 250 SC against *D. pinodes* and Toledo Extra 430 SC against *F. avenaceum* were used. Means of 3 replicates. The same letters beside the values indicate statistically insignificant differences (*p* < 0.05) based on ANOVA analysis and Tukey’s post hoc corrections (valid in rows).

**Table 4 ijms-25-04525-t004:** qPCR primers and probes used for the identification of *F. avenaceum* and *D. pinodes*.

Genotype/Gene	Primer/Probe	Sequence (5′-3′)	Regression Equation, Efficiency of qPCR (E)	References
*D. pinodes* ITS	Forward	5′-AGAGACCGATAGCGCACAAG-3′	y = −3.77x + 23.9 R^2^ = 0.96; E = 91.9	[13]
	Reverse	5′-AGTCCAGGCTGGTTGCAGGA-3′		
	Probe	FAM—CATGTACCTCTCTTCGGG—MGB		
*F. avenaceum* *Esyn1*	Forward	5′-AGCAGTCGAGTTCGTCAACAGA-3′	y = −3.44x + 19.7 R^2^ = 0.99; E = 95.3	[119]
	Reverse	5′-GGCYTTTCCTGCGAACTTG-3′		
	Probe	FAM—CCGTCGAGTCCTCT—MGB		

## Data Availability

The data presented in this study are available in this article and the Appendix A.

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
