# Peer review of "Antifungal Properties of Bio-AgNPs against D. pinodes and F. avenaceum Infection of Pea (Pisum sativum L.) Seedlings"

_ijms, 2024, doi:10.3390/ijms25084525_

Round 1

Reviewer 1 Report

Comments and Suggestions for Authors

The manuscript by Karolina Stałanowska et al entitled ‘Antifungal properties of bio-AgNPs against D. pinodes and F. avenaceum infection of pea (Pisum sativum L.) seedlings’ was reviewed. In my opinion, the manuscript is written very well, the text needs some proofreading, but some conclusions are questionable. I think the manuscript cannot be published in its current state. My main comments are listed below.

line 2-3 «F. avenaceum» in different lines, please use non-breaking space (Ctrl + Shift + Space)

line 148 «…to investigate mycelial growth inhibition.». You put resazurin assay with spores, don't you? And here you are exploring spore germination, not mycelial growth.

line 152-156 Why is resazurin assay made with spores? Spore germination and mycelium proliferation are different processes from the point of view of the physiology of the fungus cell. And since you are determining the concentration here, it is worth checking the toxic effect of nanoparticles on the plant itself.

line 170-175 line 170-175 Judging by the linear mycelium growth inhibition test (Fig. 2), nanoparticles, unlike even a fungicide, do not allow D. pinodes to begin sporulation (or greatly delay this stage). It would be great to supplement the article from this point of view by studying the effect of nanoparticles on D. pinodes in a little more detail, and then, perhaps, it will be possible to repeat a similar effect for F. avenaceum, which will be a very big plus for your work.

The petri dishes with a light background (control and treatment with fungicides) should be photographed on a dark background for contrast.

line 219-232 It would be more correct to wash off plants and calculate CFU to determine the number of viable cells. Since you are working with fungi, your DNA counting results can be influenced by quite a lot of different physiological factors of pathogen fungal cells: starting from the stage of the cell cycle of the biomass of fungi and ending with the state of the cell wall. And all this can be influenced by both fungicides and nanoparticles, as well as the process of pathogenesis itself (its presence /absence / stage). Not to mention the general problem of isolating DNA from fungi, which is a very time-consuming process (I see in the materials and methods that they were isolated according to a good protocol).

Thus, you got a figure that is rather mediocre related to the pathogenesis itself. As can be seen in Fig 5, where your disease index has a correlation with DNA R < 0.95. However, this effect can also be explained by the negative effect of silver nanoparticles on the plant itself. Where is the control with the treatment of the plant with nanoparticles, but without infection? And the same control is desirable for the fungicides used to obtain the most reliable data.

Besides, since you're giving the amount of DNA, why are you giving it by volume? The volume of what? The volume of plant tissues? The volume of the sample you obtained? In this form, the data is of little use for any conclusions..

line 197-200 The description of Table2 is either misleading and needs to be rewritten, or explain in detail why plants with different growth/infection periods were taken for measurement.

Again, there is not enough control with nanoparticle treatment and without infection, it seems that the treatment makes the plants more viable by themselves and regardless of pathogenesis.

line 214-215 You cannot claim about "shortening of stem and underdevelopment of stems and tendrils" without providing any measurement data or at least without the scale bar in Fig 4B, where you refer.

line 233-238 Fig 5B статистическая ошибка больше самих данных не есть хорошо.

Fig 7 and Table 3 - it is necessary to provide data on the treatment with nanoparticles without infection and fungicides without infection.

Having data on the amount of fungal DNA relative to plant tissues or the amount of plant DNA, it would be possible to conclude about the effect of fungal cells on the results obtained on metabolites (perhaps some of these changes relate to the contents of fungal cells, rather than peas or pathogenesis in general, or are mostly related to treatment with a fungicide or nanoparticles).

line 355-359 Are the results related to pathogens? Or just with the processing? Or with the killing of other microflora on the plant?

line 365-366 If you knew that "Fungal spores are usually more sensitive to nanoparticles than mycelium hyphae", then why did resazurin assay determine the minimum concentration on spores, and not on hyphae?

line 381 You are talking about comparing the exposure time and the concentration of nanoparticles, translate the ppm concentration into line with other values discussed for ease of perception.

line 393-394 And only here do you mention that you infected plants with a spore preparation. It is worth adding this to the description of resazurin assay, so that it is clear why the minimum concentration is determined in this way.

line 402 and 404 If you compare concentrations, either ppm everywhere or mg/L everywhere for easier perception.

line 407-408 In your Table2, there is data where a plant treated with 200mg/L nanoparticles gives results more than the control plant. And you say "these nanoparticles did not affect the growth and development of the seedlings". Either repeat the analysis of the table, or reformulate the table so that the "statistically significant (P<0.05) differences" are clearly and unambiguously indicated.

line 435-436 «pea-F. avenaceum» it needs to be edited, the extra hyphen and the species name should be in italics.

line 472 and 472 «D. pinodes» the D in both cases isn’t italics, please correct.

line 492-493 «F. avenaceum» in different lines, please use non-breaking space (Ctrl + Shift + Space)

Please replace “ca” with “approximately” everywhere in text. It’s hard to understand for non-native English speakers, and there are plenty of them among researches around the globe.

line 545-546 «F. avenaceum» in different lines, please use non-breaking space (Ctrl + Shift + Space)

line 576-577 Here, perhaps, it would be possible to draw a conclusion in favor of one hypothesis or another, if you had control with nanoparticle treatment, but without infection.

line 606-613 In this case, it is necessary to study the effect of Lactobacillus paracasei metabolites separately from silver, perhaps it is in them. Or expand on this topic in more detail with links to other articles, if such data already exists.

line 623-624 Can you provide some links to different papers to explain why you are shaken them so long? Are you sure they didn’t start the germination? Or provide microphotograph of the suspension after shaking, to ensure there is no germination yet. If they are started sprout – your data on MIC (on spores) and experiment design with pathogenesis are not match each other.

line 627-628 Why are CFU concentrations different for the determination of MIC and plant infection?

line 643 «The plates were incubated for 4 days at 22°C (day/night, 12h/12h).» Did the plates shake? If not, then during incubation at the bottom of the wells the concentration of both spores and nanoparticles may increase significantly.

line 619 and 656 Decide whether "petri dishes" or "Petri dishes"

line 664-672 Were the plants not grown sterile? In this case, many of the effects of nanoparticles on plants can be explained by non-specific suppression of microflora. It is also worth considering that the development of pathogenesis in such conditions could be significantly influenced by opportunistic microorganisms.

line 688 How exactly you inoculated the pathogens? Drop few suspension drips on the different places of plants? Pulverize them? Place entire part of plant into suspension of spores?

line 696-697 If you just throw the seedlings in a liquid nitrogen – your quantitative DNA data is completely useless. Some of the fungal cells may be just washed with the nitrogen. Can you provide more detailed information about storing samples for further qPCR?

line 712 While you take only 200mg of samples, you will get drastically statistical error.

line 719-720 «D. pinodes» in different lines, please use non-breaking space (Ctrl + Shift + Space)

In general, the article is good and with interesting results. It is necessary to correct a few small errors in the text, provide additional information on the methods and clarify a couple of points

Comments on the Quality of English Language

Author Response

Reviewer 1

Dear reviewer,

We would like to express our sincere gratitude for your valuable feedback on the manuscript. Your comments have been a great source of inspiration, and we appreciate your efforts in improving the quality of our work. We have carefully analyzed all your comments and responded to each of them below. Thank you once again for your time and contribution.

Best regards.

Question 1: “line 2-3 «F. avenaceum» in different lines, please use non-breaking space (Ctrl + Shift + Space)” AND “line 435-436 «pea-F. avenaceum» it needs to be edited, the extra hyphen and the species name should be in italics.” AND “line 472 and 472 «D. pinodes» the D in both cases isn’t italics, please correct.” AND “line 492-493 «F. avenaceum» in different lines, please use non-breaking space (Ctrl + Shift + Space)” AND “Please replace “ca” with “approximately” everywhere in text. It’s hard to understand for non-native English speakers, and there are plenty of them among researches around the globe.” AND “line 545-546 «F. avenaceum» in different lines, please use non-breaking space (Ctrl + Shift + Space)” AND “line 719-720 «D. pinodes» in different lines, please use non-breaking space (Ctrl + Shift + Space)” AND “line 381 You are talking about comparing the exposure time and the concentration of nanoparticles, translate the ppm concentration into line with other values discussed for ease of perception.” AND “line 402 and 404 If you compare concentrations, either ppm everywhere or mg/L everywhere for easier perception.” AND “line 619 and 656 Decide whether "petri dishes" or "Petri dishes"”.

Answer 1: Thank you for pointing out the typographical errors and all the editorial comments. This has been corrected accordingly.

Q2: “line 148 «…to investigate mycelial growth inhibition.». You put resazurin assay with spores, don't you? And here you are exploring spore germination, not mycelial growth.”

A2: Resazurin assay was performed using fungal spores. Therefore, we have clarified this in the text – Line 148-151: “The antifungal properties of bio-AgNPs against D. pinodes and F. avenaceum were evaluated by a resazurin assay to determine the minimum inhibitory concentration (MIC) of bio-AgNPs against fungal spores and by the poisoned media technique to investigate mycelial growth inhibition.”

Q3: “line 152-156 Why is resazurin assay made with spores? Spore germination and mycelium proliferation are different processes from the point of view of the physiology of the fungus cell. And since you are determining the concentration here, it is worth checking the toxic effect of nanoparticles on the plant itself.”

A3: Resazurin assay is performed to investigate the viability of eukaryotic cells. For fungal cell viability, both hyphae [Chadha and Kale 2015; Abd Algaffar et al. 2021; Silva et al. 2023] and spores [Monteiro et al. 2012; Barua et al. 2017; Herman et al. 2024] are used. We wanted to investigate whether the nanoparticles would have a negative effect on both mycelium growth and spore viability – spores as the infection agent and mycelium as established fungal colony.

Abd Algaffar, S.O.; Verbon, A.; van de Sande, W.W.J.; Khalid, S.A. Development and validation of an in vitro resazurin-based susceptibility assay against madurella mycetomatis. Antimicrob. Agents Chemother. 2021, 65, 14–17, doi:10.1128/AAC.01338-20.

Barua, P.; You, M.P.; Bayliss, K.; Lanoiselet, V.; Barbetti, M.J. A rapid and miniaturized system using Alamar blue to assess fungal spore viability: implications for biosecurity. Eur. J. Plant Pathol. 2017, 148, 139–150, doi:10.1007/s10658-016-1077-5.

Chadha, S.; Kale, S.P. Simple fluorescence-based high throughput cell viability assay for filamentous fungi. Lett. Appl. Microbiol. 2015, 61, 238–244, doi:10.1111/lam.12460.

Herman, T.S.; da Silva Goersch, C.; Bocca, A.L.; Fernandes, L. Resazurin to determine the minimum inhibitory concentration on antifungal susceptibility assays for Fonsecaea sp. using a modified EUCAST protocol. Brazilian J. Microbiol. 2024, doi:10.1007/s42770-024-01293-2.

Monteiro, M.C.; De La Cruz, M.; Cantizani, J.; Moreno, C.; Tormo, J.R.; Mellado, E.; De Lucas, J.R.; Asensio, F.; Valiante, V.; Brakhage, A.A.; et al. A new approach to drug discovery: High-throughput screening of microbial natural extracts against Aspergillus fumigatus using resazurin. J. Biomol. Screen. 2012, 17, 542–549, doi:10.1177/1087057111433459.

Silva, T.C.; Moreira, S.I.; Assis, F.G.; Vicentini, S.N.C.; Silva, A.G.; Oliveira, T.Y.K.; Christiano, F.S.; Custódio, A.A.P.; Leite, R.P.; Gasparoto, M.C.G.; et al. An accurate, affordable, and precise resazurin-based digital imaging colorimetric assay for the assessment of fungicide sensitivity status of fungal populations. Agronomy 2023, 13, 1–19, doi:10.3390/agronomy13020343.

Q4: “Thus, you got a figure that is rather mediocre related to the pathogenesis itself. As can be seen in Fig 5, where your disease index has a correlation with DNA R < 0.95. However, this effect can also be explained by the negative effect of silver nanoparticles on the plant itself. Where is the control with the treatment of the plant with nanoparticles, but without infection? And the same control is desirable for the fungicides used to obtain the most reliable data.” AND “Again, there is not enough control with nanoparticle treatment and without infection, it seems that the treatment makes the plants more viable by themselves and regardless of pathogenesis.” AND “Fig 7 and Table 3 - it is necessary to provide data on the treatment with nanoparticles without infection and fungicides without infection.” AND “line 576-577 Here, perhaps, it would be possible to draw a conclusion in favor of one hypothesis or another, if you had control with nanoparticle treatment, but without infection.”.

A4: This study aimed to evaluate the antifungal properties and effectiveness of bio-AgNPs against D. pinodes and F. avenaceum, in vitro and in planta. We focused on the effect of nanoparticle usage on limiting infection in comparison with the effect of fungicides. We did not investigate the effects of the nanoparticles on plants. Our results suggest that nanoparticles reduce infection at a similar level as fungicides, and that they affect the further growth of pea seedlings in a similar way. Fungicides have been approved for use in pea protection in EU. Therefore, we did not consider their possible negative effects on seedling development. Moreover, our unpublished data suggests that pea exposure to bio-AgNPs does not negatively affect plant growth.

Q5: “line 170-175 Judging by the linear mycelium growth inhibition test (Fig. 2), nanoparticles, unlike even a fungicide, do not allow D. pinodes to begin sporulation (or greatly delay this stage). It would be great to supplement the article from this point of view by studying the effect of nanoparticles on D. pinodes in a little more detail, and then, perhaps, it will be possible to repeat a similar effect for F. avenaceum, which will be a very big plus for your work.”

A5: Thank you very much for this inspiring suggestion. In our next research we intend to determine the activity of AgNPs against other important pea pathogens such as P. medicaginis and F. oxysporum, so we will use this suggestion.

Q6: “The petri dishes with a light background (control and treatment with fungicides) should be photographed on a dark background for contrast.”

A6: We are aware of the inconvenience. When preparing the photographic documentation, it seemed that the mycelium was clearly visible in the photographs. Subsequent photo processing showed that the mycelium (especially D. pinodes control) could be more visible. Thank you for your suggestion regarding the dark background. We will use it in subsequent photos.

Q7: “line 219-232 It would be more correct to wash off plants and calculate CFU to determine the number of viable cells. Since you are working with fungi, your DNA counting results can be influenced by quite a lot of different physiological factors of pathogen fungal cells: starting from the stage of the cell cycle of the biomass of fungi and ending with the state of the cell wall. And all this can be influenced by both fungicides and nanoparticles, as well as the process of pathogenesis itself (its presence /absence / stage). Not to mention the general problem of isolating DNA from fungi, which is a very time-consuming process (I see in the materials and methods that they were isolated according to a good protocol).”

A7: The methodology used in this study has been previously validated in other studies, and we are confident of its accuracy. For the initial step of DNA isolation, infected plant tissues were ground in liquid nitrogen. This method has been proven to be the most effective way to break down both plant and fungal cells, as demonstrated in our studies conducted on both pure fungal cultures and plant tissues. All experimental variants were treated in the same manner, which we believe minimizes the potential for experimental error.

Q8: “Besides, since you're giving the amount of DNA, why are you giving it by volume? The volume of what? The volume of plant tissues? The volume of the sample you obtained? In this form, the data is of little use for any conclusions.”

A8: Thank you very much for catching this oversight. This error must have occurred while creating the graphs and an incorrect unit was mistakenly inserted. The data were checked again, and the amount of DNA was expressed in pg, but the unit was incorrectly described as pg/µL. This has been corrected in the figures and the text of the manuscript.

Q9: “line 197-200 The description of Table2 is either misleading and needs to be rewritten, or explain in detail why plants with different growth/infection periods were taken for measurement.”

A9: Our intention was to underline that pre-treatment with water, fungicide, and bio-AgNPs was performed on the 7th day of germination, and plants were infected the next day – on the 8th day of germination. Such a detailed description can be misleading, so this has been corrected – line 199-201: “The fresh weight (FW) and length of shoot and root of 22-day-old pea seedlings developed without infection (control), infected (on the 8th day of germination, DG) with D. pinodes or F. avenaceum after pre-treatment with water, fungicide and bio-AgNPs (at 100 and 200 mg/L).”

Q10: “line 214-215 You cannot claim about "shortening of stem and underdevelopment of stems and tendrils" without providing any measurement data or at least without the scale bar in Fig 4B, where you refer.”

A10: Thank you for your valuable comment. This fragment refers not only to the measurement results, but also to the observations of seedlings, whose appearance can be seen in the figures. According to your suggestion, we have added references to Table 2, where are the seedling measurements.

Line 215-216: “(…) shortening of stem and underdevelopment of stipules and tendrils (Table 2, Figure 4B). (…)”

Q11: “line 233-238 Fig 5B статистическая ошибка больше самих данных не есть хорошо” ; translation - “Statistical error larger than the data itself is not good”.

A11: We believe that the advantage of our research work we have planned is that it complements classical phytopathological studies with the use of a molecular technique that allows the assessment of infection levels through DNA quantification of both pathogens (TaqMan qPCR). The assessment of pea plant health using a disease index revealed some discrepancies in the level of infection in different replicates of the same research variants. However, the statistical analysis showed that the results obtained by both research methods corresponded well with each other, as shown in Figure 6. Based on our study, it can be concluded that classical phytopathological analyses should be supported by molecular methods which are more precise.

Q12: “Having data on the amount of fungal DNA relative to plant tissues or the amount of plant DNA, it would be possible to conclude about the effect of fungal cells on the results obtained on metabolites (perhaps some of these changes relate to the contents of fungal cells, rather than peas or pathogenesis in general, or are mostly related to treatment with a fungicide or nanoparticles).”

A12: The ratio of the amount of fungal DNA to total isolated DNA was small. Approximately 35 µg of total DNA was isolated from 200 mg of the tissue sample. The largest quantity of fungal DNA present in the sample was approximately 500 pg, which was approximately 0.0014% of the total DNA. Consequently, the quantity of pathogen cells in the plant tissues was negligible, and therefore, this aspect was deemed insignificant and was not discussed.

Q13: “line 355-359 Are the results related to pathogens? Or just with the processing? Or with the killing of other microflora on the plant?”

A13: As described in the beginning of paragraph “3.1. Antifungal activity of bio-AgNPs” we discussed the antifungal properties of bio-AgNPs against D. pinodes and F. avenaceum. However, we agree that the title could be clarified. Therefore, changed in line 357: “3.1. Antifungal activity of bio-AgNPs against D. pinodes and F. avenaceum

Q14: “line 365-366 If you knew that "Fungal spores are usually more sensitive to nanoparticles than mycelium hyphae", then why did resazurin assay determine the minimum concentration on spores, and not on hyphae?”

A14: Spores are typically more sensitive than hyphae but not always. For example, Tarazona et al. (2019) showed that for F. poae, the effective doses (ED50, ED90, and ED100) of AgNPs are very similar (even identical for 2 and 30 h of exposure) between spores and mycelium, regardless of the time of exposure. Similar results were also obtained for F. graminearum, F. sporotrichoides, and F. langsethiae, especially for 30h of exposure. In addition, there is no information about the sensitivity of D. pinodes and F. avenaceum to bio-AgNPs, which was important considering that we used spores to infect plants.

Tarazona, A.; Gómez, J. V.; Mateo, E.M.; Jiménez, M.; Mateo, F. Antifungal effect of engineered silver nanoparticles on phytopathogenic and toxigenic Fusarium spp. and their impact on mycotoxin accumulation. Int. J. Food Microbiol. 2019, 306, 108259, doi:10.1016/j.ijfoodmicro.2019.108259

Q15: “line 393-394 And only here do you mention that you infected plants with a spore preparation. It is worth adding this to the description of resazurin assay, so that it is clear why the minimum concentration is determined in this way.”

A15: Thank you for this comment. We agree that this may be misleading. Information that the MIC was for fungal spores was added as mentioned in Q2 and in line 691-692: “To determine the minimum inhibitory concentration (MIC) for fungal spores the resazurin assay was performed (…)”.

Q16: “line 407-408 In your Table2, there is data where a plant treated with 200mg/L nanoparticles gives results more than the control plant. And you say "these nanoparticles did not affect the growth and development of the seedlings". Either repeat the analysis of the table, or reformulate the table so that the "statistically significant (P<0.05) differences" are clearly and unambiguously indicated.”

A16: This is especially true in the case of roots of pea seedlings infected with D. pinodes after treatment with bio-AgNPs at a concentration of 200 mg/L. The authors meant that there was no negative impact of nanoparticles, so this fragment was corrected in line 441-442: “(…) these nanoparticles did not negatively affect the growth and development of the seedlings (Tables 2 and S1).”

Q17: “line 606-613 In this case, it is necessary to study the effect of Lactobacillus paracasei metabolites separately from silver, perhaps it is in them. Or expand on this topic in more detail with links to other articles, if such data already exists.”

A17: This is an interesting aspect that needs to be explored. However, our previous work suggests that this may not have an impact. Lahuta et al. 2022 and 2023 investigated the effect of bio-AgNPs on wheat seedlings. Nanoparticles showed toxic effects on wheat seedlings at concentrations of 20-80 mg/L – much lower than in the case of this study for pea. Moreover, our unpublished data suggests that pea exposure to bio-AgNPs does not negatively affect plant growth. Therefore, bio-AgNPs seem to have a more plant species-specific effect than Lactobacillus paracasei metabolites on nanoparticles.

Lahuta, L.B.; Szablińska-Piernik, J.; Głowacka, K.; Stałanowska, K.; Railean-Plugaru, V.; Horbowicz, M.; Pomastowski, P.; Buszewski, B. The effect of bio-synthesized silver nanoparticles on germination, early seedling development, and metabolome of wheat (Triticum aestivum L.). Molecules 2022, 27, 2303, doi:10.3390/molecules27072303.

Lahuta, L.B.; Szablińska-Piernik, J.; Stałanowska, K.; Horbowicz, M.; Górecki, R.J.; Railean, V.; Pomastowski, P.; Buszewski, B. Exogenously applied cyclitols and biosynthesized silver nanoparticles affect the soluble carbohydrate profiles of wheat (Triticum aestivum L.) seedling. Plants 2023, 12, 1627, doi:10.3390/plants12081627.

Q18: “line 623-624 Can you provide some links to different papers to explain why you are shaken them so long? Are you sure they didn’t start the germination? Or provide microphotograph of the suspension after shaking, to ensure there is no germination yet. If they are started sprout – your data on MIC (on spores) and experiment design with pathogenesis are not match each other.”

A18: Thank you for pointing out this oversight. In the original draft of the publication, the preparation of spore suspensions was described separately for MIC and plant infections. Later, the descriptions were combined and the fact that the spores for the resazurin assay were used immediately after collection was overlooked. In the case of plant infection, the spores were shaken for 24 h to initiate germination and increase the chance of infection. It was corrected – line 684-686: “The suspension was transferred to a 25 mL glass bottle and use immediately in resazurin assay or shaken for 24 h at 22°C and 110 rpm and use for plant infection.”

Q19: “line 627-628 Why are CFU concentrations different for the determination of MIC and plant infection?”

A19: CFU concentrations were higher for plant infection to make the inoculation more successful, considering that some spores might not be transferred but remain on the sterile synthetic brush used for precise inoculation.

Q20: “line 643 «The plates were incubated for 4 days at 22°C (day/night, 12h/12h).» Did the plates shake? If not, then during incubation at the bottom of the wells the concentration of both spores and nanoparticles may increase significantly.”

A20:  The plates were not shaken. When adapting the method, we did not find any protocol with shaking (for example: Robles‐Martínez et al. 2019; Athira et al. 2021). This is a limitation of the proposed method. However, even the use of shaking would not improve the dispersion of both nanoparticles and spores, which would fall to the bottom anyway. This screening method allowed us to select the concentration range of bio-AgNPs, despite its drawbacks.

Athira, K.; Gurrala, L.; Kumar, D.V.R. Biosurfactant-mediated biosynthesis of CuO nanoparticles and their antimicrobial activity. Appl. Nanosci. 2021, 11, 1447–1457, doi:10.1007/s13204-021-01766-y.

Robles‐Martínez, M.; González, J.F.C.; Pérez‐Vázquez, F.J.; Montejano‐Carrizales, J.M.; Pérez, E.; Patiño‐Herrera, R. Antimycotic activity potentiation of Allium sativum extract and silver. Chem. Biodivers. 2019, 16, 1–14, doi:10.1002/cbdv.201800525.

Q21: “line 664-672 Were the plants not grown sterile? In this case, many of the effects of nanoparticles on plants can be explained by non-specific suppression of microflora. It is also worth considering that the development of pathogenesis in such conditions could be significantly influenced by opportunistic microorganisms.”

A21: All seedlings were incubated under the same conditions in a previously sterilized incubator and placed in sterile tubes with sterilized water. Yes, the seedlings were not sterilized to avoid disturbing their natural physiology (and also natural microflora as part of it) and to not provide another stressor that could affect their response to infection.

Q22: „line 688 How exactly you inoculated the pathogens? Drop few suspension drips on the different places of plants? Pulverize them? Place entire part of plant into suspension of spores?”

A22: Seedlings were inoculated with a sterile synthetic brush, which was dipped in spore suspension, and then on the plants, seedlings infected with D. pinodes were inoculated on the shoot and root bases, whereas those infected with F. avenaceum were inoculated on the base of the shoot and root (as presented below in Figure 1).

Q23: “line 696-697 If you just throw the seedlings in a liquid nitrogen – your quantitative DNA data is completely useless. Some of the fungal cells may be just washed with the nitrogen. Can you provide more detailed information about storing samples for further qPCR” AND “line 712 While you take only 200mg of samples, you will get drastically statistical error.”

A23: Seedlings from each treatment were placed in aluminum foil envelopes and then placed in liquid nitrogen. The envelopes were tightly wrapped, so there was no way to wash the fungal cells from the plant surface. In addition, fungal cells were not only present on the plant surface, but also in the plant tissues. Samples were stored in an ultra-freezer at -80°C. The next day, the seedlings were ground in liquid nitrogen with a mortar and pestle and placed in eppendorf tubes. Samples (200 mg) of homogenized material from three replicates were used for DNA isolation. The rest of the plant material was stored in an ultra-freezer at -80°C. The isolated gDNA was stored at 4°C until the next day for qPCR.

Reviewer 2 Report

Comments and Suggestions for Authors

The manuscript by Stałanowska et al. reveals exciting results that suggest a potential application of bio-AgNPs against D. pinodes and F. gallinaceum infection of peas. The results of the biological tests, both in vitro and in vivo, are satisfactory, and I consider that they should be published. However, the authors must provide complete information corresponding to the characterization of said bio-AgNPs, even if they have already been published. It can be done in the supplementary information document. I request it since various instrumental techniques characterize silver nanoparticles that allow studying their morphology, size, distribution, crystalline structure, chemical composition, and optical properties, among other characteristics. Among the techniques used are transmission electron microscopy (TEM), which allows observing the morphology and size of nanoparticles at the nanometric level; scanning electron microscopy (SEM), which provides high-resolution images of the surface of the nanoparticles, atomic absorption spectroscopy (AAS), which is used to determine the concentration of silver in the nanoparticles and provides information on the chemical composition and its purity; Ultraviolet-visible absorption spectroscopy (UV-Vis), which allows the optical properties of nanoparticles to be characterized, such as absorbance and light scattering; X-ray diffraction (XRD), which is used to determine the crystal structure of silver nanoparticles, and allows the different crystal planes to be identified and the crystallite size to be calculated; Raman spectroscopy that provides information on the chemical composition and molecular structure of silver nanoparticles, and allows studying the molecular vibrations and excitation modes of the particles and dynamic particle size analysis (DLS), used to determine the hydrodynamic size and size distribution of nanoparticles in suspension. The authors indicate that the nanoparticles were obtained following reference 56, in which only Electron Microscopy, Dynamic Light Scattering Analysis, Fluorescence Spectroscopy, and Fourier Transform Infrared Spectroscopic Analysis were used to achieve their characterization. From my chemical point of view, it is not sufficient information.

It is desirable, then, that the authors use some of the techniques suggested to complement said characterization, giving parameters in the experimental section such as acceleration voltage, type of contrast, mode of operation (high resolution (HRTEM) or conventional mode), and exposure time for TEM. X-ray diffraction should also be used, reporting the angle of incidence and DLS and selecting the appropriate angle to obtain the best resolution. XPS is also suggested to establish the excitation energy and AASS to reveal the silver concentration in the nanoparticles.

Once the nanoparticle characterization process is completed, the manuscript will be of higher quality and should be published.

Author Response

Dear reviewer,

We would like to express our sincere gratitude for your valuable feedback on the manuscript. Your comments have been a great source of inspiration, and we appreciate your efforts in improving the quality of our work. We have carefully analyzed all your comments and responded to each of them below. Thank you once again for your time and contribution.

Best regards.

Q1: The manuscript by Stałanowska et al. reveals exciting results that suggest a potential application of bio-AgNPs against D. pinodes and F. gallinaceum infection of peas. The results of the biological tests, both in vitro and in vivo, are satisfactory, and I consider that they should be published. However, the authors must provide complete information corresponding to the characterization of said bio-AgNPs, even if they have already been published. It can be done in the supplementary information document. I request it since various instrumental techniques characterize silver nanoparticles that allow studying their morphology, size, distribution, crystalline structure, chemical composition, and optical properties, among other characteristics. Among the techniques used are transmission electron microscopy (TEM), which allows observing the morphology and size of nanoparticles at the nanometric level; scanning electron microscopy (SEM), which provides high-resolution images of the surface of the nanoparticles, atomic absorption spectroscopy (AAS), which is used to determine the concentration of silver in the nanoparticles and provides information on the chemical composition and its purity; Ultraviolet-visible absorption spectroscopy (UV-Vis), which allows the optical properties of nanoparticles to be characterized, such as absorbance and light scattering; X-ray diffraction (XRD), which is used to determine the crystal structure of silver nanoparticles, and allows the different crystal planes to be identified and the crystallite size to be calculated; Raman spectroscopy that provides information on the chemical composition and molecular structure of silver nanoparticles, and allows studying the molecular vibrations and excitation modes of the particles and dynamic particle size analysis (DLS), used to determine the hydrodynamic size and size distribution of nanoparticles in suspension. The authors indicate that the nanoparticles were obtained following reference 56, in which only Electron Microscopy, Dynamic Light Scattering Analysis, Fluorescence Spectroscopy, and Fourier Transform Infrared Spectroscopic Analysis were used to achieve their characterization. From my chemical point of view, it is not sufficient information. It is desirable, then, that the authors use some of the techniques suggested to complement said characterization, giving parameters in the experimental section such as acceleration voltage, type of contrast, mode of operation (high resolution (HRTEM) or conventional mode), and exposure time for TEM. X-ray diffraction should also be used, reporting the angle of incidence and DLS and selecting the appropriate angle to obtain the best resolution. XPS is also suggested to establish the excitation energy and AASS to reveal the silver concentration in the nanoparticles.

Once the nanoparticle characterization process is completed, the manuscript will be of higher quality and should be published.

A1: Thank you for your encouraging remarks and for highlighting the importance of a comprehensive characterization of the bio-silver nanoparticles (bio-AgNPs) presented in our study. We recognize the necessity of detailing the characterization techniques to substantiate our findings effectively. In response to your suggestions, we have expanded the revised manuscript of the characterization methods used in our research. This includes additional information on the Scanning and Transmission Electron Microscopy equipped with Energy Dispersive X-ray Spectroscopy that we utilized to analyze the morphology and elemental composition of the nanoparticles. Furthermore, we have employed X-ray Photoelectron Spectroscopy to determine the surface chemistry of silver in bio-AgNPs, documenting the excitation energy and other relevant conditions. Details on these methods have been added to both the supplementary materials (Figure S2) and the revised sections of the methodology (line 650-673) and results discussion (line 361-393) in the revised manuscript.

We have also incorporated results from Inductively Coupled Plasma Optical Emission Spectrometry to quantify the concentration of silver, ensuring a comprehensive understanding of the chemical composition of the bio-AgNPs. We trust that these amendments will address your concerns and enhance the manuscript's suitability for publication.

Round 2

Reviewer 2 Report

Comments and Suggestions for Authors

The authors have changed the manuscript, improving its quality and providing responses to the observations and comments. Therefore, I suggest the publication of the current version of the manuscript.